# BUILDING TRANSFORMATION LAYERS FOR RIEMANNIAN NEURAL NETWORKS

## ABSTRACT

Recently, deep neural networks on manifold-valued representations have garnered significant attention in various machine learning applications. One recent focus is to generalize Euclidean Fully Connected (FC) and convolutional layers to non-Euclidean geometries. However, previous approaches typically focus on a few selected manifolds and rely on the specific properties of the target manifold. In contrast, this work proposes a framework for constructing FC and convolutional layers over computationally tractable Riemannian spaces, using only Riemannian geometry. This framework incorporates several previous FC layers across different geometries as special cases, and is instantiated over ten representative manifolds, including three hyperbolic models, five geometries of the Symmetric Positive Definite (SPD) manifold, and two Grassmannian perspectives. Experiments on different manifolds demonstrate the effectiveness and applicability of our approach.

## 1 INTRODUCTION

Deep neural networks on Riemannian manifolds have achieved remarkable success across various applications (Huang and Van Gool, 2017; Ganea et al., 2018; Skopek et al., 2020; López et al., 2021; Huang et al., 2022; Wang et al., 2024b; Chen and Lipman, 2024; Khan et al., 2025; Pouliquen et al., 2025; Li et al., 2025). Commonly encountered manifolds include Symmetric Positive Definite (SPD) (Pennec et al., 2006), Grassmannian (Bendokat et al., 2024), matrix Lie groups Hall (2013), and hyperbolic (Ungar, 2022a) manifolds. These manifolds admit *computationally tractable tools* such as geodesics, exponential and logarithmic maps, and parallel transport, which have enabled the extension of fundamental deep learning components, including normalization (Brooks et al., 2019; Chakraborty, 2020; Lou et al., 2020; Kobler et al., 2022; Chen et al., 2024b; 2025a; Wang et al., 2025b), attention (Gulcehre et al., 2019; Pan et al., 2022; Wang et al., 2024a; 2025a), residual blocks (van Spengler et al., 2023; Katsman et al., 2024; He et al., 2025), and classification layers (Ganea et al., 2018; Nguyen and Yang, 2023; Chen et al., 2024a;c; Bdeir et al., 2024).

Yet, the generalization of the most fundamental layers, Fully Connected (FC) and convolutional layers, remains particularly challenging. Early works targeted specific manifolds: Huang and Van Gool (2017); Huang et al. (2017; 2018) proposed layers for SPD, special orthogonal, and Grassmann manifolds. Later, Ganea et al. (2018); Mao et al. (2024) developed hyperbolic counterparts via tangent spaces, and Chen et al. (2022) introduced the Lorentz linear layer in spacetime. To better respect geometry, Shimizu et al. (2021) extended the FC and convolutional layers on the Poincaré model, while Nguyen et al. (2024; 2025) proposed SPD counterparts based on gyrovector structure and invariant metric on symmetric spaces. However, these methods largely rely on specific properties (e.g., Poincaré geometry, gyro structures, or symmetric structures), which restricts their generality. In another direction, Chakraborty et al. (2020) introduced a convolution based on the weighted Fréchet mean, but unlike the Euclidean convolution, its output manifold dimension is restricted to match the input, limiting flexibility. Consequently, a general and flexible framework for constructing FC and convolutional layers across different geometries remains unsolved.

We address this challenge by proposing a principled framework for building Riemannian FC and convolutional layers on computationally tractable manifolds. Our framework relies solely on Riemannian operators such as exponential and logarithmic maps. It thus applies broadly to different geometries, such as hyperbolic, SPD, and Grassmannian spaces. Our contributions are summarized as follows.

- **Riemannian FC and convolutional layers.** We introduce a principled generalization of FC and convolutional layers to Riemannian spaces. In contrast to previous approaches, our framework only requires tractable Riemannian operators, ensuring broad applicability. Moreover, several existing Riemannian FC layers are subsumed as special cases.
- **Ten concrete instantiations.** We instantiate our framework on three hyperbolic models, five SPD geometries, and two Grassmannian perspectives. Our approach enables direct variation of the latent geometry under a consistent network architecture.
- **Empirical validation.** We validate our approach on benchmark tasks across hyperbolic, SPD, and Grassmannian manifolds, demonstrating both effectiveness and versatility.

## 2 PRELIMINARIES

**Notations.** For the Euclidean space $\mathbb{R}^n$ or $\mathbb{R}^{n \times n}$, we denote $\langle \cdot, \cdot \rangle$ as the standard inner product, with $\|\cdot\|$ as the induced norm, *i.e.*, $L_2$-norm for vectors and Frobenius norm for matrices. A Riemannian manifold $(\mathcal{M}, g)$ with the Riemannian metric $g$ is abbreviated as $\mathcal{M}$. Its tangent space at $P \in \mathcal{M}$ is denoted as $T_P\mathcal{M}$. The Riemannian logarithm, exponentiation, and metric at $P \in \mathcal{M}$ are denoted as $\text{Log}_P$, $\text{Exp}_P$, and $\langle \cdot, \cdot \rangle_P = g_P(\cdot, \cdot)$, respectively. The parallel transport along the geodesic connecting $P, Q \in \mathcal{M}$ is $\Gamma_{P \to Q}$. Besides, a complete table of notation is summarized in Sec. C.

**Hyperbolic manifold.** There are five isometric hyperbolic models (Cannon et al., 1997). We focus on the Poincaré ball $\mathbb{P}^n_K = \left\{ x \in \mathbb{R}^n \mid \|x\|^2 < -1/K \right\}$, Beltrami–Klein ball $\mathbb{K}^n_K = \left\{ x \in \mathbb{R}^n \mid \|x\|^2 < -1/K \right\}$, and hyperboloid (or Lorentz) $\mathbb{H}^n_K = \left\{ x \in \mathbb{R}^{n+1} \mid \|x\|^2_{\mathcal{L}} = 1/K, x_1 > 0 \right\}$, where $\|x\|^2_{\mathcal{L}} = \sum_{i=2}^{n+1} x_i^2 - x_1^2$ is the Lorentz inner product. Here, $K < 0$ is the constant curvature. The Poincaré and Beltrami–Klein ball admit gyrovector spaces, known as the Möbius and Einstein gyrovector spaces (Ungar, 2022b), respectively. The Möbius gyroaddition and scalar gyromultiplication are denoted as $\oplus_M$ and $\otimes_M$, while the Einstein counterparts are $\oplus_E$ and $\otimes_E$. Sec. D.1 summarizes the associated Riemannian and gyro operators.

**SPD manifold.** The set of $n \times n$ SPD matrices, denoted $\mathcal{S}^n_{++}$, forms a smooth manifold, called the SPD manifold (Arsigny et al., 2005). It admits five widely used Riemannian metrics: Affine-Invariant Metric (AIM) (Pennec et al., 2006), Log-Euclidean Metric (LEM) (Arsigny et al., 2005), Power-Euclidean Metric (PEM) (Dryden et al., 2010), Log-Cholesky Metric (LCM) (Lin, 2019), and Bures–Wasserstein Metric (BWM) (Bhatia et al., 2019). Each of them provides closed-form expressions for Riemannian operators, summarized in Sec. D.2.

**Grassmannian.** The Grassmannian is the manifold of $p$-dimensional subspaces of the $n$-dimensional vector space (Tu, 2011, Prob. 7.8). It has two common matrix representations (Bendokat et al., 2024). The Projector Perspective (PP) embeds each element as an $n \times n$ symmetric matrix: $\widetilde{\text{Gr}}(p, n) = \{P \in \mathcal{S}^n | P^2 = P, \text{rank}(P) = p\}$, where $\mathcal{S}^n$ is the Euclidean space of symmetric matrices. The OrthoNormal Basis (ONB) perspective is the quotient of the Stiefel manifold $\text{St}(p, n)$: $\text{Gr}(p, n) = \text{St}(p, n)/\text{O}(p) = \{[U] | [U] = \{\widetilde{U} \in \text{St}(p, n) \mid \widetilde{U} = UR, R \in \text{O}(p)\}\}$, where $\text{O}(p)$ is the $p \times p$ orthogonal group. By abuse of notation, we use $[U]$ and $U$. Their Riemannian structures are summarized in Sec. D.3.

The considered manifolds admit multiple geometries, including isometric hyperbolic models, distinct SPD metrics, and diffeomorphic Grassmannian perspectives, whose empirical performance often varies across tasks (Nguyen, 2022; Katsman et al., 2024; Chen et al., 2025b). This motivates a unified framework to flexibly handle such variants. Besides, exponential and logarithmic maps may be singular, but can be numerically solved (Rmks. D.1 and D.2), and are assumed well-defined.

## 3 PROPOSED FRAMEWORK

### 3.1 RIEMANNIAN FULLY CONNECTED LAYERS

Our method for building FC layers over the Riemannian manifold relies on point-to-hyperplane distance, which has shown success in building hyperbolic and SPD networks (Shimizu et al., 2021; Nguyen and Yang, 2023; Bdeir et al., 2024; Chen et al., 2024a;c; Nguyen et al., 2024; 2025). The

hyperplane in the Riemannian manifold $\mathcal{N}$ (Chen et al., 2024c, Eq. 5) is $H_{A,P} = \{X \in \mathcal{N} : \langle \text{Log}_P(X), A \rangle_P = 0\}$, with $P \in \mathcal{N}$ and $A \in T_P\mathcal{N}$. When $\mathcal{N} = \mathbb{R}^n$, it recovers the Euclidean hyperplane, $H_{a,p} = \{x \in \mathbb{R}^n : \langle a, x - p \rangle = 0\}$.

The Euclidean FC layer is defined as $y = Ax + b$ with $A \in \mathbb{R}^{m \times n}$ and $b \in \mathbb{R}^m$. It can be expressed element-wise as $y_k = \langle a_k, x \rangle - b_k = \langle a_k, x - p_k \rangle$ with $a_k, p_k \in \mathbb{R}^n$ and $\langle p_k, a_k \rangle = b_k$. As shown by Shimizu et al. (2021, Sec. 3.2), the LHS $y_k$ is the signed distance of $y$ to the hyperplane passing through the origin and orthogonal to the $k$-th axis of the output space, which can be formulated as

$$\text{sign}(\langle e_k, y - \mathbf{0} \rangle) d(y, H_{e_k, \mathbf{0}}) = \langle a_k, x - p_k \rangle, \quad \forall 1 \leq k \leq m, \tag{1}$$

where $\mathbf{0} \in \mathbb{R}^m$ is the zero vector, and $\{e_k\}_{k=1}^m$ forms an orthonormal basis over $\mathbb{R}^m$ with $e_k$ denoting the vector whose $k$-th element is 1 and all others are 0. Here, the LHS of Eq. (1) equals $y_k$.

Given the point-to-hyperplane distance, Eq. (1) can be readily generalized into the manifold. Noting that $\text{Log}_p(x) = x - p$ under the Euclidean geometry and that $T_{\mathbf{0}}\mathbb{R}^m \cong \mathbb{R}^m$, the counterparts of $H_{e_k, \mathbf{0}}$ on an $m$-dimensional Riemannian manifold $\mathcal{M}$ are defined as

$$H_{B_k, E} = \{S \in \mathcal{M} : \langle \text{Log}_E S, B_k \rangle_E = 0\}, \quad \forall 1 \leq k \leq m, \tag{2}$$

where $E \in \mathcal{M}$ is the predefined origin, and $\{B_k\}_{k=1}^m$ is an orthogonal basis over $\{T_E\mathcal{M}, g_E\}$. Eq. (2) characterizes the hyperplane containing the origin $E$ and orthogonal to the geodesic starting from $E$ with the initial velocity $B_k$, which recovers the Euclidean $H_{e_k, \mathbf{0}}$ as $\mathcal{M} = \mathbb{R}^m$. With all the above ingredients, we define the Riemannian FC layer in the following.

**Definition 3.1.** Given $n$-dimensional manifold $\mathcal{N}$ and $m$-dimensional manifold $\mathcal{M}$, the Riemannian FC layer $\mathcal{F} : \mathcal{N} \to \mathcal{M}$ for the input $X \in \mathcal{N}$ returns the output $Y \in \mathcal{M}$ by solving $m$ equations:

$$\text{sign}\left(\langle \text{Log}_E^{\mathcal{M}}(Y), B_k \rangle_E^{\mathcal{M}}\right) \text{d}^{\mathcal{M}}(Y, H_{B_k, E^{\mathcal{M}}}^{\mathcal{M}}) = \langle A_k, \text{Log}_{P_k}^{\mathcal{N}}(X) \rangle_{P_k}^{\mathcal{N}}, 1 \leq k \leq m, \tag{3}$$

where $E^{\mathcal{M}} \in \mathcal{M}$ is the origin, $\{B_k\}_{k=1}^m$ is an orthonormal basis over $T_{E^{\mathcal{M}}}\mathcal{M}$. Here, $\text{d}^{\mathcal{M}}$ is the point-to-hyperplane distance over $\mathcal{M}$, while $\text{Log}_{P_i}^{\mathcal{N}}$ and $\langle \cdot, \cdot \rangle_{P_i}^{\mathcal{N}}$ are the Riemannian logarithm and metric over $\mathcal{N}$. Each $P_k \in \mathcal{N}$ and $A_k \in T_{P_k}\mathcal{N}$ are the FC parameters.

Our Def. 3.1 naturally extends previous FC layers over different geometries.

**Proposition 3.2.** [↓] *When $\mathcal{N} = \mathbb{R}^n$ and $\mathcal{M} = \mathbb{R}^m$, Def. 3.1 reduces to the Euclidean FC layer. When $\mathcal{N} = \mathbb{P}_K^n$, $\mathcal{M} = \mathbb{P}_K^m$, the point-to-hyperplane distance follows Ganea et al. (2018, Thm. 5), and the LHS of Eq. (3) follows $v_k(\cdot)$ by Shimizu et al. (2021, Eq. (3)), Def. 3.1 yields the Poincaré FC layer (Shimizu et al., 2021, Sec. 3.2). When $\mathcal{N} = \mathcal{S}_{++}^n$, $\mathcal{M} = \mathcal{S}_{++}^m$, and the point-to-hyperplane distances are pseudo-gyrodistances (Nguyen and Yang, 2023, Thms. 2.23–2.25), Def. 3.1 recovers the corresponding gyro SPD FC layers (Nguyen et al., 2024, Props. 3.4–3.6).*

The crux of Def. 3.1 lies in the point-to-hyperplane distance and solving the resulting $m$ equations. The previous work typically requires a case-by-case derivation for a specific geometry. However, Chen et al. (2024c, Thm. 3.2) recently introduced a Riemannian point-to-hyperplane distance based on Riemannian trigonometry: $d(X, H_{A_k, P_k}) = \frac{|\langle \text{Log}_{P_k}(X), A_k \rangle_{P_k}|}{\|A_k\|_{P_k}}$, where $\|\cdot\|_{P_k}$ denotes the norm induced by $\langle \cdot, \cdot \rangle_{P_k}$. Under this distance, the implicit Def. 3.1 admits an explicit solution.

**Theorem 3.3** (Riemannian FC Layers). [↓] *Following the notation in Def. 3.1, the Riemannian FC layer $\mathcal{F}(\cdot) : \mathcal{N} \to \mathcal{M}$ for the input $X \in \mathcal{N}$ is $Y = \text{Exp}_E^{\mathcal{M}}\left(\sum_{i=1}^m \langle \text{Log}_{P_i}^{\mathcal{N}}(X), A_i \rangle_{P_i}^{\mathcal{N}} B_i\right)$, where $\text{Exp}_E^{\mathcal{M}}$ is the Riemannian exponentiation over $\mathcal{M}$.*

Our FC layer differs from the tangent FC layer by a single tangent space (Ganea et al., 2018, Lem. 6), which is $\text{Exp}_E(f(\text{Log}_E(X)))$ with $f$ as a Euclidean FC layer. In contrast, our formulation involves multiple tangent spaces, where each $\langle \text{Log}^{\mathcal{N}} P_i(X), A_i \rangle_{P_i}^{\mathcal{N}} = 0$ corresponds to a Riemannian hyperplane $H_{A_i, P_i}$. Besides, our formulation naturally generalizes prior Riemannian FC layers without requiring additional geometric or algebraic structures, such as Poincaré geometry (Shimizu et al., 2021), gyrovector spaces (Nguyen et al., 2024), and symmetric spaces (Nguyen et al., 2025).

## 3.2 RIEMANNIAN CONVOLUTIONAL LAYERS

As shown by Shimizu et al. (2021, Sec. 3.4), the Euclidean convolution takes the FC transformation on each receptive field. Let us focus on a single receptive field. Given a $c$-channel concatenated

feature vector $x \in (\mathbb{R}^n)^c$ in a receptive field, the $k$-th output of this receptive field can be described as an affine transformation, $y_k = \langle a_k, x \rangle - b_k$. Therefore, the Riemannian convolution can be defined by the Riemannian FC layer discussed in Sec. 3.1 within each receptive field.

**Riemannian convolution.** In a receptive field, the manifold-valued features $\{X_i \in \mathcal{N}\}_{i=1}^c$ are first concatenated into $X \in (\mathcal{N})^c$, which is then fed into $k$ Riemannian FC layers, where $k$ is the number of kernels. Here, each Riemannian FC layer is implemented under the product geometry $(\mathcal{N})^c = \Pi_{i=1}^c \mathcal{N}$, which is detailed in Sec. E.3. Fig. 1 illustrates the above process.

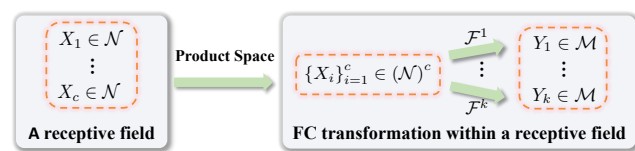

Figure 1: Riemannian convolution within a receptive field. Here, $\mathcal{F}^k(\cdot)$ denotes the $k$-th FC transformation.

### 3.3 PARAMETERS TRIVIALIZATION

As convolution takes the FC layer as the prototype, we focus on the FC parameters. Since $P_i$ varies during training, $A_i \in T_{P_i}\mathcal{N}$ cannot be updated directly by the Euclidean optimizer. As shown by Chen et al. (2024c, Eqs. (12)-(13)), it can be determined from the fixed tangent space at the origin $E^{\mathcal{N}} \in \mathcal{N}$ by[1] $A_i = \Gamma_{E^{\mathcal{N}} \to P_i}(Z_i)$ with $Z_i \in T_{E^{\mathcal{N}}}\mathcal{N}$. Besides, as shown by Shimizu et al. (2021, Sec. 3.1), $p_k$ might be overly parameterized, as there are countless $p_k$ satisfying $\langle a_k, p_k \rangle = b_k$. Therefore, following Shimizu et al. (2021), each $P_i$ in the Riemannian FC layer is parameterized as $\text{Exp}_{E^{\mathcal{M}}}^{\mathcal{M}}(\gamma_i[Z_i])$, where $\gamma_i \in \mathbb{R}$ and $[Z_i]$ is the unit vector of $Z_i$. In this way, all FC parameters can be directly optimized by a Euclidean optimizer. Note that optimizing manifold-valued parameters by the exponential map is known as trivialization (Lezcano Casado, 2019, Sec. 4.1).

## 4 EXAMPLES

Although Thm. 3.3 is geometry-agnostic, its instantiation can be further simplified under a specific geometry. We now manifest our FC layer in Thm. 3.3 over different geometries, including three hyperbolic models, five SPD geometries, and two Grassmannian perspectives.

### 4.1 HYPERBOLIC VECTOR MANIFOLDS

Table 1: Comparison of hyperbolic FC layers. An extended table can be found in Sec. F.

| Method | Model | Mechanism | References |
|---|---|---|---|
| Möbius | $\mathbb{P}_K^n$ | Tangent | (Ganea et al., 2018, Def. 3.2) |
| Einstein | $\mathbb{K}_K^n$ | Tangent | (Mao et al., 2024, Thm. 9) |
| Lorentz | $\mathbb{H}_K^n$ | Spacetime | (Chen et al., 2022, Sec 3.1) |
| Poincaré FC | $\mathbb{P}_K^n$ | Poincaré | (Shimizu et al., 2021, Sec. 3.2) |
| Ours | $\mathbb{P}_K^n, \mathbb{K}_K^n, \mathbb{H}_K^n$ | Riemannian | Thms. 4.1 and 4.2 |

We focus on three hyperbolic models: Poincaré ball, Beltrami–Klein, and hyperboloid models. The resulting FC layers are denoted as HFC-P, HFC-K, and HFC-H, respectively. Tab. 1 compares our hyperbolic FC layer against previous ones.

**Poincaré & Beltrami–Klein.** These two models admit Möbius and Einstein gyrovector spaces (Ungar, 2022b), respectively. These structures further simplify the concrete HFC layers.

**Theorem 4.1** (HFC-P & HFC-K). [↓] *Let $\mathcal{H}^n \in \{\mathbb{P}_K^n, \mathbb{K}_K^n\}$. Given $x \in \mathcal{H}^n$, the Riemannian FC layer $\mathcal{F}(\cdot) : \mathcal{H}^n \to \mathcal{H}^m$ is $y = \text{Exp}_{\mathbf{0}}\left((v_1(x), \cdots, v_m(x))^\top\right)$. Here, $v_i(x) = \langle \text{Log}_{\mathbf{0}}(-p_i \oplus_{\mathcal{H}} x), z_i \rangle$, with the zero vector $\mathbf{0}$ as the origin and $p_i = \text{Exp}_{\mathbf{0}}(\gamma_i[z_i])$. The FC parameters are $\{\gamma_i \in \mathbb{R}\}_{i=1}^m$ and $\{z_i \in \mathbb{R}^n\}_{i=1}^m$. The gyroaddition and Riemannian exp and log can be found in Sec. D.1, where $\text{Exp}_{\mathbf{0}}$ ($\text{Log}_{\mathbf{0}}$) shares the same expression in the two models.*

Interestingly, the only difference between the HFC-P and HFC-K layers lies in the gyroaddition. Besides, HFC-P takes a different expression from the Poincaré FC layer (Shimizu et al., 2021, Sec. 3.2), as their point-to-hyperplane distances and LHSs of Eq. (3) are different.

**Hyperboloid.** The origin is defined as $e = (1/\sqrt{|K|}, 0, \cdots, 0)^\top$, which corresponds to the Poincaré origin under the stereographic projection (Skopek et al., 2020, Sec. 2.1). Then, we have the following.

---

[1] Although $\Gamma$ could be flexibly replaced by other maps between tangent spaces, such as vector transport and differential of group translation, we mainly use parallel transport.

**Theorem 4.2** (HFC-H FC layer). [↓] *The Riemannian FC layer $\mathcal{F}(\cdot) : \mathbb{H}_K^n \to \mathbb{H}_K^m$ for $x \in \mathbb{H}_K^n$ is $y = \mathrm{Exp}_e\left((0, v_1(x), \cdots, v_m(x))^\top\right)$, where $v_i(x) = \left\langle \mathrm{Log}_{p_i}(x), \Gamma_{e \to p_i}(z_i) \right\rangle$, and $p_i = \mathrm{Exp}_e(\gamma_i[(0, z_i^\top)^\top])$, with $\gamma_i \in \mathbb{R}$ and $z_i \in \mathbb{R}^n$ as parameters.*

## 4.2 SPD MATRIX MANIFOLDS

We focus on five popular Riemannian metrics, *i.e.*, LEM, AIM, PEM, LCM, and BWM. We define the identity matrix $I$ as the origin, as it corresponds to the zero matrix under the matrix logarithm.

**Theorem 4.3** (SPD FC Layers). [↓] *Given an SPD matrix $S \in \mathcal{S}_{++}^n$, the outputs of the SPD FC layers $\mathcal{F}(\cdot) : \mathcal{S}_{++}^n \to \mathcal{S}_{++}^m$ under different Riemannian metrics are*

$$LEM : Y = \exp\left(V^{\mathrm{LE}}\right), V_{ij}^{\mathrm{LE}} = \begin{cases} \frac{1}{\sqrt{\alpha}} v_{ii}^{\mathrm{LE}}(S) + \mu \sum_{k=1}^m v_{kk}^{\mathrm{LE}}(S), & \text{if } i = j \\ \frac{1}{\sqrt{2\alpha}} v_{ij}^{\mathrm{LE}}(S), & \text{if } i > j \\ V_{ji}^{\mathrm{LE}}, & \text{otherwise} \end{cases} \tag{4}$$

$$AIM : Y = \exp\left(V^{\mathrm{AI}}\right), V_{ij}^{\mathrm{AI}} = \begin{cases} \frac{1}{\sqrt{\alpha}} v_{ii}^{\mathrm{AI}}(S) + \mu \sum_{k=1}^m v_{kk}^{\mathrm{AI}}(S), & \text{if } i = j \\ \frac{1}{\sqrt{2\alpha}} v_{ij}^{\mathrm{AI}}(S), & \text{if } i > j \\ V_{ji}^{\mathrm{AI}}, & \text{otherwise} \end{cases} \tag{5}$$

$$PEM : Y = \left(I + V^{\mathrm{PE}}\right)^{\frac{1}{\theta}}, V_{ij}^{\mathrm{PE}} = \begin{cases} \frac{1}{\sqrt{\alpha}} v_{ii}^{\mathrm{PE}}(S) + \mu \sum_{k=1}^m v_{kk}^{\mathrm{PE}}(S), & \text{if } i = j \\ \frac{1}{\sqrt{2\alpha}} v_{ij}^{\mathrm{PE}}(S), & \text{if } i > j \\ V_{ji}^{\mathrm{PE}}, & \text{otherwise} \end{cases} \tag{6}$$

$$LCM : Y = V^{\mathrm{LC}}(V^{\mathrm{LC}})^\top, V_{ij}^{\mathrm{LC}} = \begin{cases} \exp\left(v_{ii}^{\mathrm{LC}}(S)\right), & \text{if } i = j \\ v_{ij}^{\mathrm{LC}}(S), & \text{if } i > j \\ 0, & \text{otherwise} \end{cases} \tag{7}$$

$$BWM : Y = \left(I + \frac{1}{2} V^{\mathrm{BW}}\right)^2, V_{ij}^{\mathrm{BW}} = \begin{cases} v_{ii}^{\mathrm{BW}}(S), & \text{if } i = j \\ \frac{1}{\sqrt{2}} v_{ij}^{\mathrm{BW}}(S), & \text{if } i > j \\ V_{ji}^{\mathrm{BW}}, & \text{otherwise} \end{cases} \tag{8}$$

*Here, $v_{ij}(S)$ under different metrics are*

$$LEM : \left\langle \log(S) - \log(P_{ij}), Z_{ij} \right\rangle^{(\alpha,\beta)}, \quad AIM : \left\langle \log(P_{ij}^{-\frac{1}{2}} S P_{ij}^{-\frac{1}{2}}), Z_{ij} \right\rangle^{(\alpha,\beta)},$$

$$PEM : \left\langle S^\theta - P_{ij}^\theta, Z_{ij} \right\rangle^{(\alpha,\beta)}, \quad LCM : \left\langle \lfloor K \rfloor - \lfloor L_{ij} \rfloor + \mathrm{Dlog}(\mathbb{KL}_{ij}^{-1}), \lfloor Z_{ij} \rfloor + \frac{1}{2}\mathbb{Z}_{ij} \right\rangle,$$

$$BWM : \left\langle (P_{ij}S)^{\frac{1}{2}} + (SP_{ij})^{\frac{1}{2}} - 2P_{ij}, \mathcal{L}_{P_{ij}}(L_{ij} Z_{ij} L_{ij}^\top) \right\rangle,$$

*The above notations are defined in the following.*

- *$Z_{ij} \in T_I \mathcal{S}_{++}^n \cong \mathcal{S}^n$ and $P_{ij} \in \mathcal{S}_{++}^n$ are the parameters for $1 \le i \le j \le m$,*
- *$\log(\cdot)$ is the matrix logarithm. $\mathrm{Dlog}(\cdot)$ is the diagonal element-wise logarithm. $\lfloor \cdot \rfloor$ is the strictly lower part of a square matrix. $\mathrm{Chol}(\cdot)$ is the Cholesky decomposition. $\mathbb{V}$ is a diagonal matrix with diagonal elements of the square matrix $V$. $\mathcal{L}_P(V)$ is the solution to the matrix linear system $\mathcal{L}_P[V]P + P\mathcal{L}_P[V] = V$, known as the Lyapunov operator. $\mu = \frac{1}{n}\left(\frac{1}{\sqrt{\alpha+n\beta}} - \frac{1}{\sqrt{\alpha}}\right)$, $K = \mathrm{Chol}(S)$ and $L_{ij} = \mathrm{Chol}(P_{ij})$.*
- *$\langle \cdot, \cdot \rangle$ and $\langle \cdot, \cdot \rangle^{(\alpha,\beta)}$ are the Frobenius inner product and $\mathrm{O}(n)$-invariant one defined in Eq. (26).*
- *Due to the incompleteness of PEM and BWM, there are constraints for $V^{\mathrm{PE}}$ and $V^{\mathrm{BW}}$: $I + \theta V^{\mathrm{PE}} \in \mathcal{S}_{++}^m$ and $I + \frac{1}{2} V^{\mathrm{BW}} \in \mathcal{S}_{++}^n$. Both constraints can be solved by numerical regularization, as detailed in Rmk. G.4.*

The Euclidean affine FC $y = Ax + b$ incorporates the linear map $y = Ax$, the most natural map between linear spaces. As shown by Arsigny et al. (2005, Sec. 4.4) and Chen et al. (2024d, Thm. 1), the SPD manifold admits two vector space structures w.r.t. LEM and LCM. Similar to the Euclidean FC layer, our SPD FC layer also incorporates linear maps over these vector structures. Denoting the addition and scalar product as $\oplus^{\mathrm{LE}}$ ($\oplus^{\mathrm{LC}}$) and $\odot^{\mathrm{LE}}$ ($\odot^{\mathrm{LC}}$), which are detailed in Sec. J.6, we have the following result.

**Proposition 4.4.** [↓] *The LEM- and LCM-SPD FC layers incorporate the linear homomorphisms over the vector spaces $\{\mathcal{S}_{++}^n, \oplus^{\text{LE}}, \odot^{\text{LE}}\}$ and $\{\mathcal{S}_{++}^n, \oplus^{\text{LC}}, \odot^{\text{LC}}\}$, respectively.*

**Comparison.** As summarized in Tab. 2, three gyro SPD FC layers Nguyen et al. (2024, Props. 3.4-3.6) and two flat SPD FC layers (Nguyen et al., 2025) are incorporated by our SPD FC layers.

Table 2: Comparison with the existing SPD FC layers.

| SPD FC Layer | Geometries | Requirement | Incorporated by Ours |
|---|---|---|---|
| Gyro FC (Nguyen et al., 2024) | AIM, LEM & LCM | Gyrovector | ✓(Sec. G.1) |
| Flat FC (Nguyen et al., 2025) | LEM & LCM | Flat geometry | ✓(Sec. G.2) |
| Symmetric FC (Nguyen et al., 2025) | AIM | Invariant metric Symmetric space | N/A |
| Ours | Riemannian spaces | Riemannian | N/A |

**Simplification and convolution.** Following the trivialization in Sec. 3.3, the SPD FC layers under LEM, AIM, LCM, and PEM can be further simplified, as detailed in Sec. G.3. The convolution is defined as Sec. 3.2, with $\mathcal{M} = \mathcal{S}_{++}^m$ and $\mathcal{N} = \mathcal{S}_{++}^n$.

### 4.3 GRASSMANNIAN MATRIX MANIFOLDS

We manifest our FC layers over the ONB and PP Grassmannian, and define Grassmannian Convolution (GrConv) as Sec. 3.2, Then, we compare our GrConv with existing popular Grassmannian transformations, concluding that our GrConv are more flexible on both dimensionality and geometries.

**ONB.** We denote $I_{p,n} = [I_p, \mathbf{0}]^\top \in \mathbb{R}^{n \times p}$, with $I_p$ as the $p \times p$ identity matrix. We define it as the Grassmannian origin, as it corresponds to $I_n \in \mathrm{O}(n)$ in the quotient structure (Bendokat et al., 2024, Sec. 2.2). As Sec. 3.3, the FC parameters are modeled by parallel transport and Riemannian exponential map at $I_{p,n}$. The concrete ONB Grassmannian FC layer can be further simplified.

**Theorem 4.5** (ONB). [↓] *Given $U \in \mathrm{Gr}(p,n)$, the ONB Grassmannian FC layer $\mathcal{F}(\cdot)$ : $\mathrm{Gr}(p,n) \to \mathrm{Gr}(q,m)$ is $Y = \begin{pmatrix} R\cos(\Sigma)R^\top \\ O\sin(\Sigma)R^\top \end{pmatrix}$, with $B^{\text{ONB}} \overset{SVD}{:=} O\Sigma R^\top \in \mathbb{R}^{(m-q)\times q}$.*

*Each $(i,j)$ element of $B^{\text{ONB}} \in \mathbb{R}^{(m-q)\times q}$ is $\left\langle \mathrm{Log}_{P_{ij}}^{\text{ONB}}(U), T_{ij}B_{Z_{ij}} \right\rangle$, with $T_{ij} = \begin{pmatrix} -R_{ij}\sin(\Sigma_{ij})O_{ij}^\top \\ O_{ij}\cos(\Sigma_{ij})O_{ij}^\top + I_{n-p} - O_{ij}O_{ij}^\top \end{pmatrix}$. Here, $\gamma_{ij}[B_{Z_{ij}}] \overset{SVD}{:=} O_{ij}\Sigma_{ij}R_{ij}^\top$ is the SVD decomposition. The FC parameters are $B_{Z_{ij}} \in \mathbb{R}^{(n-p)\times p}$ and $\gamma_{ij} \in \mathbb{R}$ for $1 \le i \le m-q$ and $1, \le j \le, q$.*

**PP.** We define the PP origin as $\widetilde{I}_{p,n} = I_{p,n}I_{p,n}^\top$, as it corresponds to $I_{p,n}$ (Bendokat et al., 2024, Eq. 2.11). Similarly, we model the FC parameters by parallel transport and Riemannian exponential map at $\widetilde{I}_{p,n}$. The PP Grassmannian can be further simplified. Besides, the Riemannian logarithm under the PP Grassmannian can be calculated by the ONB logarithm to support the auto-differentiation (Nguyen et al., 2024, Prop. 3.12). For more details, please refer to the proof of the following theorem.

**Theorem 4.6** (PP). [↓] *Given $X \in \widetilde{\mathrm{Gr}}(p,n)$, the PP Grassmannian FC layer $\mathcal{F}(\cdot)$ : $\widetilde{\mathrm{Gr}}(p,n) \to \widetilde{\mathrm{Gr}}(q,m)$ is $Y = \widetilde{U}\widetilde{U}^\top$, with $\widetilde{U} = \left( \exp\left( \begin{pmatrix} 0 & -(B^{\text{PP}})^T \\ B^{\text{PP}} & 0 \end{pmatrix} \right) \right)_{1:q}$, where $(\cdot)_{1:q}$ returns the first-$q$ columns of the input square matrix. Each $(i,j)$ element of $B^{\text{PP}} \in \mathbb{R}^{(m-q)\times q}$ is defined as $\frac{1}{2}\left\langle \pi_{*,\pi(P)}\left( \mathrm{Log}_{(O_{ij})_{1:p}}^{\text{ONB}}(\pi^{-1}(X)) \right), O_{ij}Z_{ij}O_{ij}^\top \right\rangle$, with $O_{ij} = \exp\left( \begin{pmatrix} 0 & -(\gamma_{ij}[B_{Z_{ij}}])^T \\ \gamma_{ij}[B_{Z_{ij}}] & 0 \end{pmatrix} \right)$, where $\pi(U) = UU^\top$, and $\pi_{*,U}(V) = UV^\top + VU^\top$ is the differential map for all $U \in \mathrm{Gr}(p,n)$ and $V \in T_U\mathrm{Gr}(p,n)$. The FC parameters are $B_{Z_{ij}} \in \mathbb{R}^{(n-p)\times p}$ and $\gamma_{ij} \in \mathbb{R}$ for $1 \le i \le m-q$ and $1, \le j \le, q$.*

**Comparison.** Huang et al. (2018) proposed FRMap + ReOrth layers to perform the transformation over the ONB Grassmannian via left matrix product (FRMap) and QR decomposition (ReOrth). Nguyen (2022) proposed the PP scaling for the PP Grassmannian by the tangent space at the identity. Nguyen and Yang (2023) extended the PP scaling to the ONB Grassmannian. Besides, Nguyen and Yang (2023) used the gyrogroup left translation (GrTrans) as the transformation. These layers are briefly recapped in Sec. H. However, the previous layers fail to faithfully respect the Grassmannian geometries and lack flexibility regarding dimensions and perspectives. In contrast, given a $c$-channel Grassmannian $\mathrm{Gr}(p,n)$ (or $\widetilde{\mathrm{Gr}}(p,n)$) input,

our GrConv can adjust all dimensions across both perspectives, enabling more flexibility. Tab. 3 summarizes the above discussion.

### 4.4 MANIFOLD EMBEDDING

In several applications (Chami et al., 2019; López et al., 2021; Zhao et al., 2023; Nguyen et al., 2024), Euclidean feature are embedded into the manifold via $\mathrm{Exp}_E(Ax + b)$. As detailed in

Table 3: Comparison of our GrConv against the existing Grassmannian transformation layers.

| Methods | Perspective | Flexible dimensions | | |
|---|---|---|---|---|
| | | Subspace $p$ | Ambient $n$ | Channel |
| FRMap + ReOrth (Huang et al., 2018, Eqs. (2-4)) | ONB | ✗ | ✓ | ✗ |
| PP Scaling (Nguyen, 2022, Sec. 4.2.2) | PP | ✗ | ✗ | ✗ |
| ONB Scaling (Nguyen and Yang, 2023, Sec. 3.2) | ONB | ✗ | ✗ | ✗ |
| GrTrans (Nguyen and Yang, 2023, Sec. 2.3.2) | ONB + PP | ✗ | ✗ | ✗ |
| GrConv | ONB + PP | ✓ | ✓ | ✓ |

Sec. E.4, our framework implies that this operation respects the Riemannian FC layer between the Euclidean space and the target manifold, *i.e.*, $\mathcal{F}(\cdot) : \mathbb{R}^n \to \mathcal{M}$.

## 5 EXPERIMENTS

We evaluate the effectiveness of our layers on different manifolds. We refer the reader to Secs. I.1 to I.3 for experimental details on the hyperbolic, SPD, and Grassmannian spaces, respectively.

### 5.1 EXPERIMENTS ON THE HYPERBOLIC MANIFOLD

We compare our HFC layers against other hyperbolic transformation layers, including Möbius (Ganea et al., 2018) and Einstein (Mao et al., 2024) transformations via the tangent space, Lorentz linear layer (Chen et al., 2022) via the spacetime, and Poincaré FC layer (Shimizu et al., 2021). Compared to previous lay-

Table 4: Comparison of hyperbolic transformations on link prediction, where $\delta$ is the graph hyperbolicity (lower is more hyperbolic). The top 3 results are highlighted with **red**, **blue** and **cyan**.

| Method | Mechanism | Geometry | Disease $\delta = 0$ | Airport $\delta = 1$ | Pubmed $\delta = 3.5$ | Cora $\delta = 11$ |
|---|---|---|---|---|---|---|
| Möbius (Ganea et al., 2018) | Tangent | Poincaré | 75.1 ± 0.3 | 90.8 ± 0.2 | **94.9 ± 0.1** | 89.0 ± 0.1 |
| Einstein (Mao et al., 2024) | Tangent | Klein | **78.7 ± 1.0** | 93.1 ± 0.2 | **95.0 ± 0.1** | 89.3 ± 0.3 |
| LorentzTan | Tangent | Hyperboloid | 75.1 ± 0.9 | 92.7 ± 0.4 | **94.99 ± 0.1** | 89.4 ± 0.6 |
| Lorentz (Chen et al., 2022) | Spacetime | Hyperboloid | 78.0 ± 0.6 | 92.4 ± 0.1 | 94.2 ± 0.1 | **91.74 ± 0.3** |
| Poincaré FC (Shimizu et al., 2021) | Riemannian | Poincaré | 77.8 ± 1.4 | 94.0 ± 0.4 | 94.3 ± 0.5 | 88.1 ± 0.3 |
| HFC-P | Riemannian | Poincaré | **81.2 ± 0.7** | **94.8 ± 0.2** | **95.0 ± 0.1** | **90.3 ± 0.2** |
| HFC-K | Riemannian | Klein | **80.2 ± 1.0** | **94.4 ± 0.4** | 94.8 ± 0.1 | 89.7 ± 0.3 |
| HFC-H | Riemannian | Hyperboloid | 77.4 ± 0.6 | **95.2 ± 0.3** | 94.0 ± 0.3 | **92.8 ± 0.1** |

ers, our methods more faithfully and flexibly respect different latent geometries. Following Chami et al. (2019), we adopt four graph datasets for the link prediction task: Disease (Anderson and May, 1991), Airport (Zhang and Chen, 2018), Pubmed (Namata et al., 2012), and Cora (Sen et al., 2008).

**Results.** Following the HNN implementation (Ganea et al., 2018; Chami et al., 2019; Mao et al., 2024), we compare different transformation layers under the backbone network with two transformation layers. Mimicking Möbius and Einstein transformation, we further implement the tangent transformation on the hyperboloid model, $\mathrm{Log}_e(M \mathrm{Log}_e(x))$, referred to LorentzTan. Tab. 4 presents the 5-fold average testing AUC results. We have the following key observations. (1) **Effectiveness:** Our HFC generally achieves superior performance against the prior hyperbolic layers. (2) **Hyperbolicity:** On datasets with low $\delta$ (*e.g.*, Disease and Airport), Riemannian transformations outperform tangent or spacetime transformations. However, on datasets with high $\delta$ (*e.g.*, Cora and Pubmed), the Riemannian performs worse or comparatively against the tangent. This trend aligns with the geometric intuition: tangent-space approximations are inherently limited in representing curved manifolds, and thus less effective in highly non-Euclidean settings. (3) **Representation power & metrics:** The optimal models vary across datasets. On Disease and Pubmed, HFC-P performs the best, while HFC-H performs the best on the other datasets. This observation underscores the importance of models in hyperbolic learning. Unlike the prior Poincaré FC layer, which is designed specifically for the Poincaré model, our Riemannian FC layer can adapt to models in a plug-and-play manner. This adaptability enhances the representation power of hyperbolic networks, making them more versatile for diverse applications.

Table 5: Comparison of hyperbolic transformations under different settings of Poincaré RResNet.

| Dataset | Num of horospheres | 50 | | | 250 | | |
|---|---|---|---|---|---|---|---|
| | Dim | 8 | 16 | 32 | 8 | 16 | 32 |
| Disease | RResNet (Katsman et al., 2024) | 76.0 ± 1.7 | 78.0 ± 2.2 | 77.4 ± 2.2 | 71.5 ± 5.1 | 78.1 ± 3.3 | 76.5 ± 2.5 |
| | Möbius+RResNet | 74.6 ± 1.9 | 74.6 ± 5.7 | 75.1 ± 2.1 | 74.0 ± 2.7 | 71.0 ± 5.2 | 73.3 ± 3.4 |
| | Poincaré FC+RResNet | 80.4 ± 0.7 | 79.1 ± 1.8 | 79.1 ± 1.6 | 80.6 ± 0.8 | 79.1 ± 0.7 | 80.1 ± 1.4 |
| | HFC-P+RResNet | **81.1 ± 0.6** | **80.0 ± 0.4** | **81.0 ± 0.6** | **80.9 ± 0.6** | **82.3 ± 0.6** | **82.1 ± 0.3** |
| Airport | RResNet (Katsman et al., 2024) | 93.4 ± 1.1 | 92.6 ± 1.1 | 93.0 ± 0.2 | 93.0 ± 0.4 | 93.0 ± 1.6 | 89.6 ± 4.7 |
| | Möbius+RResNet | 92.9 ± 0.5 | 93.0 ± 0.3 | 92.6 ± 0.3 | 92.9 ± 0.1 | 93.2 ± 0.2 | 92.9 ± 0.6 |
| | Poincaré FC+RResNet | 92.8 ± 0.6 | 93.4 ± 0.6 | 93.8 ± 0.4 | 93.5 ± 0.4 | 93.1 ± 0.4 | 93.8 ± 0.7 |
| | HFC-P+RResNet | **94.1 ± 0.5** | **93.5 ± 0.3** | **94.8 ± 0.5** | **94.1 ± 0.6** | **94.0 ± 0.4** | **94.3 ± 0.4** |
| Cora | RResNet (Katsman et al., 2024) | **86.7 ± 1.2** | 87.2 ± 1.4 | 82.4 ± 3.5 | 82.7 ± 3.0 | 84.0 ± 3.7 | 83.3 ± 1.6 |
| | Möbius+RResNet | 84.6 ± 2.9 | 86.8 ± 2.1 | 83.1 ± 2.5 | 84.1 ± 2.4 | 83.2 ± 1.6 | 83.9 ± 2.9 |
| | Poincaré FC+RResNet | 83.8 ± 2.4 | 84.6 ± 0.9 | 83.3 ± 2.7 | 82.8 ± 3.3 | 82.8 ± 3.6 | 83.3 ± 3.3 |
| | HFC-P+RResNet | 85.6 ± 0.8 | **87.6 ± 0.8** | **87.2 ± 1.8** | **87.68±1.81** | **86.08±1.72** | **86.97±1.04** |

**Ablations.** We conduct ablations on the RResNet backbone (Katsman et al., 2024). Since the hyperbolic RResNet is built on the Poincaré ball, we compare Poincaré transformation layers, *i.e.*, Möbius, Poincaré FC, and our HFC-P. In the vanilla RResNet, inputs are first projected to the target dimension using a Euclidean linear layer, followed by mapping to the hyperbolic space and processing with hyperbolic residual blocks. In contrast, we first map the input to the hyperbolic space and then apply a hyperbolic transformation layer before feeding into the residual blocks. This transformation layer can be instantiated as Möbius, Poincaré FC, or our HFC-P layer. We perform experiments across various configurations of the residual blocks, varying both the hidden dimensions and the number of horospheres. Tab. 5 presents the 5-fold average AUC results. Our HFC-P generally outperforms other hyperbolic transformations, demonstrating its effectiveness.

Table 6: Comparison of our SPDNNs against other SPD networks. The ones highlighted with ▨ are our special cases, while those marked with * are reproduced by us due to the lack of official code.

| Methods | Radar | HDM05 | FPHA | NTU60 |
|---|---|---|---|---|
| SPDNet (Huang and Van Gool, 2017) | 93.25 ± 1.10 | 64.57 ± 0.61 | 85.59 ± 0.72 | 66.36 ± 0.72 |
| SPDNetBN (Brooks et al., 2019) | 94.85 ± 0.99 | 71.28 ± 0.79 | 89.33 ± 0.49 | 69.38 ± 0.84 |
| RResNet-AIM (Katsman et al., 2024) | 95.71 ± 0.37 | 64.95 ± 0.82 | 86.63 ± 0.55 | 70.70 ± 3.81 |
| RResNet-LEM (Katsman et al., 2024) | 95.89 ± 0.86 | 70.12 ± 2.45 | 85.07 ± 0.99 | 74.67 ± 2.89 |
| SPDNetLieBN-AIM (Chen et al., 2024b) | 95.47 ± 0.90 | 71.83 ± 0.69 | 90.39 ± 0.66 | 73.34 ± 0.40 |
| SPDNetLieBN-LCM (Chen et al., 2024b) | 94.80 ± 0.71 | 71.78 ± 0.44 | 86.33 ± 0.43 | 72.54 ± 1.09 |
| SPDNetMLR (Chen et al., 2024c) | 95.64 ± 0.83 | 65.90 ± 0.93 | 85.67 ± 0.69 | 74.18 ± 1.24 |
| GyroLE* (Nguyen and Yang, 2023) | 96.24 ± 0.24 | 73.17 ± 0.37 | 90.73 ± 0.92 | 82.65 ± 0.20 |
| GyroLC* (Nguyen and Yang, 2023) | 93.60 ± 1.31 | 67.53 ± 0.85 | 76.10 ± 0.63 | 78.32 ± 0.92 |
| GyroAI* (Nguyen and Yang, 2023) | 96.29 ± 0.48 | 72.34 ± 1.06 | 89.60 ± 0.37 | 83.71 ± 0.32 |
| GyroSPD++-AIM* (Nguyen et al., 2024) | 95.20 ± 0.88 | 69.82 ± 1.79 | 89.50 ± 0.37 | 83.14 ± 0.87 |
| GyroSPD++-LEM* (Nguyen et al., 2024) | 95.04 ± 1.36 | 77.63 ± 1.01 | 88.23 ± 0.62 | **85.48 ± 1.10** |
| GyroSPD++-LCM* (Nguyen et al., 2024) | 96.24 ± 1.22 | 75.36 ± 1.08 | 81.83 ± 0.93 | 74.64 ± 2.49 |
| SPDNN-LEM | **98.27 ± 0.48** | **81.16 ± 0.93** | **91.83 ± 0.41** | **86.72 ± 0.14** |
| SPDNN-AIM | 97.63 ± 0.50 | 80.12 ± 0.78 | **91.57 ± 0.40** | 82.44 ± 0.18 |
| SPDNN-PEM | **98.43 ± 0.44** | 78.77 ± 0.45 | 90.33 ± 0.37 | 82.61 ± 0.37 |
| SPDNN-LCM | **97.65 ± 0.75** | 75.42 ± 0.95 | **91.33 ± 0.24** | 83.39 ± 0.10 |
| SPDNN-BWM | 96.40 ± 0.91 | 74.34 ± 0.86 | 90.03 ± 0.55 | **83.81 ± 0.60** |

## 5.2 EXPERIMENTS ON THE SPD MANIFOLD

Following Huang et al. (2017); Brooks et al. (2019); Katsman et al. (2024), we use the Radar dataset (Brooks et al., 2019) for radar classification, and the HDM05 (Müller et al., 2007), FPHA (Garcia-Hernando et al., 2018), and NTU60 (Shahroudy et al., 2016) datasets for human action recognition. In line with Nguyen et al. (2024), we focus on the mutual action in NTU60. Following Wang et al. (2024a); Nguyen et al. (2024), we model each sample sequence as multi-channel SPD covariance matrices of shape $[c, n, n]$.

**SPDNN.** Our SPDNN has an MLR layer (Chen et al., 2024c) stacked on top of a convolutional layer. We denote SPDNN-[Metric] as the SPDNN using convolution under the specified metric. For SPDNN-LEM, -PEM, and -LCM, the MLR is based on the same metric as the convolution. Since the MLRs under AIM and BWM are less efficient (Chen et al., 2024c), we apply LEM MLR for SPDNN-AIM and -BWM. Besides, we trivialize the SPD parameter in the MLR as Sec. 3.3, which can be further simplified (detailed in Sec. G.4). Consequently, all parameters in the SPDNNs can be optimized by a Euclidean optimizer. We compare our networks against the following SOTA SPD networks: SPDNet (Huang et al., 2017), SPDNetBN (Brooks et al., 2019), LieBN (Chen et al.,

2024b), RResNet (Katsman et al., 2024), and MLR (Chen et al., 2024c), Gyro (Nguyen and Yang, 2023), and GyroSPD++ (Nguyen et al., 2024).

**Results.** Tabs. 6 and 7 reports the 5-fold results and training efficiency, respectively. Our SPDNNs consistently outperform other SPD models. Specifically, SPDNNs exceed the classic SPDNet by up to **5.02%, 16.59%, 6.24%, and 20.36%**, respectively. Notably, SPDNN generally outperforms GyroSPD++ under LEM, LCM, and AIM in terms of both accuracy and efficiency. This advantage arises because our trivialization not only simplifies the expression of the FC and MLR layer but also mitigates the over-parameterization in GyroSPD++. In Gy-

Table 7: The average training time per epoch of our SPDNNs against GyroSPD++. A full comparison of efficiency can be found in Sec. I.2.4.

| Geometry | Method | Radar | HDM05 | FPHA | NTU60 |
|---|---|---|---|---|---|
| AIM | GyroSPD++ | 5.09 | 103.57 | 66.35 | 125.05 |
| | SPDNN | 4.84 | 101.80 | 65.42 | 124.41 |
| LEM | GyroSPD++ | 0.99 | 0.95 | 0.66 | 7.58 |
| | SPDNN | 0.86 | 0.74 | 0.63 | 5.79 |
| LCM | GyroSPD++ | 0.66 | 0.70 | 0.37 | 5.74 |
| | SPDNN | 0.65 | 0.59 | 0.35 | 3.72 |

roSPD++, each output dimension of the FC layer requires two matrix parameters, whereas our approach uses only one matrix and one scalar parameter. This reduction in parameter complexity leads to improved training efficiency and generalization. Furthermore, the variation in optimal metrics across datasets underscores the flexibility of our methods.

## 5.3 EXPERIMENTS ON THE GRASSMANNIAN

We compare our Grassmannian convolutional layer against previous transformation layers, such as FRMap + ReOrth (Huang et al., 2018), scaling (Nguyen and Yang, 2023), and GrTrans (Nguyen and Yang, 2023). Compared with the previous layers, our transformation can more faithfully respect the Grassmannian geometries while allowing greater flexibility w.r.t. dimensions and geometries. Following Nguyen and Yang (2023), each network consists of one transfor-

Table 8: Comparison of GrNNs against other Grassmannian networks on the Radar dataset. Those marked with $^*$ are reproduced by us due to the lack of official code

| Method | Subspace dims | Ambient dims | Mean±Std |
|---|---|---|---|
| GrNet (Huang et al., 2018) | 4 | 20->16 | 90.48 ± 0.76 |
| GyroGr-Scaling$^*$ (Nguyen and Yang, 2023) | 4 | 20->20 | 88.88 ± 1.52 |
| GyroGr$^*$ (Nguyen and Yang, 2023) | 4 | 20->20 | 90.64 ± 0.57 |
| GrNN-ONB | 4->4 | 20->16 | 93.92 ± 0.74 |
| | 4->4 | 20->20 | 92.83 ± 0.66 |
| | 4->6 | 20->16 | **95.23 ± 0.96** |
| | 4->8 | 20->16 | **94.77 ± 0.81** |
| GrNN-PP | 4->4 | 20->16 | 94.35 ± 0.42 |
| | 4->4 | 20->20 | **94.56 ± 0.58** |
| | 4->6 | 20->16 | 94.51 ± 0.53 |
| | 4->8 | 20->16 | 94.11 ± 0.58 |

mation layer followed by the classification. The corresponding models are denoted as GrNet (Huang et al., 2018), GyroGr-Scaling (Nguyen and Yang, 2023), GyroGr (Nguyen and Yang, 2023), GrNN-ONB, and GrNN-PP, respectively. As our GrConv allows for a more flexible change of dimensionality, we also perform ablations on subspace and ambient dimensions of the output of the FC transformation. The experiments are conducted on the Radar dataset. Following Wang et al. (2024a), we model each radar signal as a multi-channel Grassmannian tensor, *i.e.*, $[c, n, p]$ for the ONB and $[c, n, n]$ for the PP. Tab. 8 presents the 5-fold average results, demonstrating that our GrConv outperforms other Grassmannian transformation layers. Furthermore, varying the subspace dimension proves to be potentially beneficial, as our GrConv achieves the top two results under varying subspace dimensions. These results highlight the effectiveness and flexibility of our method.

## 6 CONCLUSION

This paper extends fundamental FC and convolutional layers to operate on Riemannian manifolds. Our approach offers a naturally geometry-aware generalization that is more broadly applicable than previous work. Several existing Riemannian FC layers are subsumed within our framework as special cases. Empirically, we instantiate our framework across ten different geometries, including three hyperbolic models, five SPD geometries, and two Grassmannian formulations. Extensive experiments on radar classification, human action recognition, and graph link prediction demonstrate the effectiveness and flexibility of our approach. We expect this work to facilitate further advances in deep learning on Riemannian spaces.

## REPRODUCIBILITY STATEMENT

All theoretical results are established under explicit assumptions, with complete proofs in Sec. J. The experimental details are presented in Sec. I. The code will be released upon acceptance.

## ETHICS STATEMENT

This work uses only publicly available benchmark datasets, which contain no personally identifiable or sensitive information. We do not identify ethical concerns.

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

APPENDIX CONTENTS

## LIST OF ACRONYMS

| | |
|---|---|
| ONB | OrthoNormal Basis 2 |
| PP | Projector Perspective 2 |
| | |
| FC | Fully Connected 1 |
| GrConv | Grassmannian Convolution 6 |
| | |
| AIM | Affine-Invariant Metric 2 |
| BWM | Bures–Wasserstein Metric 2 |
| LCM | Log-Cholesky Metric 2 |
| LEM | Log-Euclidean Metric 2 |
| PEM | Power-Euclidean Metric 2 |
| SPD | Symmetric Positive Definite 1 |

## A  USE OF LARGE LANGUAGE MODELS

Large Language Models (LLMs) were used primarily for language polishing and minor text editing. In limited cases, they also assisted in translating certain mathematical formulations into PyTorch code. All generated outputs were carefully reviewed and, where necessary, corrected by the authors. The authors take full responsibility for the final content of this paper.

## B  LIMITATIONS

Our framework is designed for computationally tractable Riemannian manifolds, where closed-form expressions for exponential and logarithmic maps are available. This includes many commonly used manifolds such as hyperbolic, SPD, and Grassmannian spaces. However, in cases where the underlying manifold structure is unknown or lacks tractable Riemannian operators, our approach may not be directly applicable. In such scenarios, future work could explore numerical approximations of Riemannian operators or develop new paradigms for constructing transformation layers for intractable geometry.

## C  GLOSSARY OF SYMBOLS

Tab. 10 summarizes all the notation in the main paper.

## D  GEOMETRIES OF THE INVOLVED VECTOR AND MATRIX MANIFOLDS

### D.1  GEOMETRIES OF THE HYPERBOLIC SPACE

There are five models over the hyperbolic space (Cannon et al., 1997). We focus on the Poincaré ball, Beltrami–Klein, and hyperboloid models:

$$\text{Poincaré ball: } \mathbb{P}^n_K = \left\{ x \in \mathbb{R}^n \mid \|x\|^2 < -\frac{1}{K} \right\}, \tag{9}$$

$$\text{Beltrami–Klein: } \mathbb{K}^n_K = \left\{ x \in \mathbb{R}^n \mid \|x\|^2 < -\frac{1}{K} \right\}, \tag{10}$$

$$\text{Hyperboloid: } \mathbb{H}^n_K = \left\{ x \in \mathbb{R}^{n+1} \mid \|x\|^2_{\mathcal{L}} = \frac{1}{K}, x_1 > 0 \right\}, \tag{11}$$

where $\|x\|^2_{\mathcal{L}} = \sum_{i=2}^{n+1} x_i^2 - x_1^2$ is the Lorentz inner product, and $\|\cdot\|$ is the standard $L_2$ norm induced by the standard inner product $\langle \cdot, \cdot \rangle$. Here, $K < 0$ is the constant curvature. Although the set of the Poincaré ball is identical to the Beltrami–Klein model, their Riemannian metrics are different. In fact, each of the above models has its Riemannian metric:

$$g^{\mathbb{P}}_x(v, w) = (\lambda^K_x)^2 \langle v, w \rangle, \tag{12}$$

Table 10: Summary of notation.

| Notation | Explanation |
|---|---|
| $\{\mathcal{N}, g^{\mathcal{N}}\}$ | Riemannian manifold $\mathcal{N}$ with Riemannian metric $g^{\mathcal{N}}$ |
| $\{\mathcal{M}, g^{\mathcal{M}}\}$ | Riemannian manifold $\mathcal{M}$ with Riemannian metric $g^{\mathcal{M}}$ |
| $E$ | Origin of the interested manifold |
| $T_P \mathcal{M}$ | Tangent space at $P \in \mathcal{M}$ |
| $g_p(\cdot, \cdot)$ or $\langle \cdot, \cdot \rangle_P$ | Riemannian metric at $P$ |
| $\| \cdot \|_P$ | The norm induced by $\langle \cdot, \cdot \rangle_P$ on $T_P \mathcal{M}$ |
| $\mathrm{d}(\cdot, \cdot)$ | Geodesic distance |
| $\mathrm{Log}_P$ | Riemannian logarithm at $P$ |
| $\mathrm{Exp}_P$ | Riemannian exponentiation at $P$ |
| $\Gamma_{P \to Q}$ | Parallel transport from $P$ to $Q$ along the geodesic |
| $f_{*,P}$ | Differential map of the smooth map $f$ at $P \in \mathcal{M}$ |
| $\{B_i\}_{i=1}^m$ | Standard orthonormal bases over $m$-dimensional $T_E \mathcal{M}$ |
| $\mathbb{P}_K^n, \mathbb{K}_K^n$ and $\mathbb{H}_K^n$ | Hyperbolic models of Poincaré ball, Beltrami–Klein, and hyperboloid ($K < 0$) |
| $\mathbb{R}^n$ | Euclidean space of $n$-dimensional vectors |
| $\langle \cdot, \cdot \rangle_{\mathcal{L}}$ | Lorentz inner product |
| $\oplus_{\mathrm{M}}$ and $\otimes_{\mathrm{M}}$ | Möbius gyro addition and scalar product |
| $\oplus_{\mathrm{E}}$ and $\otimes_{\mathrm{E}}$ | Einstein gyro addition and scalar product |
| $\pi_{\mathbb{K}_K^n \to \mathbb{P}_K^n}$ and $\pi_{\mathbb{P}_K^n \to \mathbb{K}_K^n}$ | Riemannian isometries between Beltrami–Klein and Poincaré ball |
| $\mathcal{S}_{++}^n$ | Space of $n \times n$ SPD matrices |
| $\mathcal{S}^n$ | Euclidean space of $n \times n$ symmetric matrices |
| $\mathcal{L}^n$ | Euclidean space of $n \times n$ lower triangular matrices |
| $\langle \cdot, \cdot \rangle$ | Standard Frobenius inner product |
| $\langle \cdot, \cdot \rangle^{(\alpha, \beta)}$ | $\mathrm{O}(n)$-invariant Euclidean metric on $\mathcal{S}^n$ s.t. $\min(\alpha, \alpha + n\beta) > 0$ |
| $\| \cdot \|_{\mathrm{F}}$ | Frobenius Norm |
| $\log$ | Matrix logarithm |
| $\exp$ | Matrix exponentiation |
| $P^{\theta}$ | Matrix power for SPD matrix $P$ |
| $\mathcal{L}_P[\cdot]$ | Lyapunov operator by $P \in \mathcal{S}_{++}^n$ |
| $\mathscr{L}$ | Cholesky decomposition |
| $\mathrm{Dlog}$ | Diagonal element-wise logarithm |
| $\lfloor \cdot \rfloor$ | Strictly lower triangular part of a square matrix |
| $\mathbb{D}(\cdot)$ | A diagonal matrix with diagonal elements from a square matrix |
| $\mathrm{Gr}(p, n)$ | Grassmannian under the ONB perspective |
| $\widetilde{\mathrm{Gr}}(p, n)$ | Grassmannian under the projector perspective |
| $\mathcal{Q}(\cdot)$ | Return an orthogonal matrix by QR decomposition |
| $[\cdot, \cdot]$ | Matrix commutator |
| $I_{p,n}$ | Grassmannian identity under the ONB perspective |
| $\widetilde{I}_{p,n}$ | Grassmannian identity under the projector perspective |
| $I_n$ | $n \times n$ identity matrix |
| $\pi$ | Riemannian isometry from $\mathrm{Gr}(p, n)$ onto $\widetilde{\mathrm{Gr}}(p, n)$ |
| $\overline{(\cdot)}$ | $\overline{(\cdot)} = \widetilde{\mathrm{Log}}_{\widetilde{I}_{p,n}}(\cdot)$ with $\widetilde{\mathrm{Log}}$ as the Riemannian logarithm on $\widetilde{\mathrm{Gr}}(p, n)$ |
| $\mathbf{0}$ | Zero matrix or vector |
| $\mathrm{St}(p, n)$ | Stiefel manifold of $n \times p$ column-wise orthogonal matrices |
| $\mathrm{GL}(n)$ | General linear group of $n \times n$ invertible matrices |
| $\mathrm{O}(n)$ | Orthogonal group of $n \times n$ orthogonal matrices |

$$g_x^{\mathbb{K}}(v, w) = \frac{\langle v, w \rangle}{1 + K \|x\|^2} - \frac{K \langle x, v \rangle \langle x, w \rangle}{\left(1 + K \|x\|^2\right)^2}, \tag{13}$$

$$g_x^{\mathbb{H}}(v, w) = \langle v, w \rangle_{\mathcal{L}} = \sum_{i=2}^{n+1} v_i w_i - v_1 w_1, \tag{14}$$

where $\lambda_x^K = \frac{2}{(1 + K \|x\|^2)}$ is a conformal factor.

As shown by Ungar (2022b), both the Poincaré ball and Beltrami–Klein models admit gyrovector structures, which are the natural counterparts of vector space in the manifold. The Poincaré ball admits a Möbius gyrovector space (Ungar, 2022b, Ch. 6.14), while the Beltrami–Klein model admits an Einstein gyrovector space (Ungar, 2022b, Ch. 6.18). Denoting $\mathcal{H} \in \{\mathbb{P}_K^n, \mathbb{K}_K^n\}$, for any $x, y \in \mathcal{H}$

Table 11: Riemannian operators on the Poincaré ball and hyperboloid ($K < 0$).

| Operator | $\mathbb{P}^n_K = \left\{ x \in \mathbb{R}^n \mid \|x\|^2 < -\frac{1}{K} \right\}$ | $\mathbb{H}^n_K = \left\{ x \in \mathbb{R}^{n+1} \mid \|x\|^2_{\mathcal{L}} = \frac{1}{K}, x_1 > 0 \right\}$, with $\|x\|^2_{\mathcal{L}} = \sum_{i=2}^{n+1} x_i^2 - x_1^2$ |
|---|---|---|
| $g_x(v,w)$ | $(\lambda_x^K)^2 \langle v, w \rangle$, $\lambda_x^K = \frac{2}{(1+K\|x\|^2)}$ | $\langle v, w \rangle_{\mathcal{L}} = \sum_{i=2}^{n+1} v_i w_i - v_1 w_1$ |
| $\mathrm{d}(x,y)$ | $\frac{2}{\sqrt{|K|}} \tanh^{-1}\left( \sqrt{|K|} \| - x \oplus_M y\| \right)$ | $\frac{1}{\sqrt{|K|}} \cosh^{-1}\left( |K| \langle x, y \rangle_{\mathcal{L}} \right)$ |
| $\mathrm{Log}_x(y)$ | $\frac{2}{\sqrt{|K|}\lambda_x^K} \tanh^{-1}\left( \sqrt{|K|} \| -x \oplus_M y\| \right) \frac{-x \oplus_M y}{\| -x \oplus_M y\|}$ | $\frac{\cosh^{-1}(K\langle x,y\rangle_{\mathcal{L}})}{\sinh\left(\cosh^{-1}(K\langle x,y\rangle_{\mathcal{L}})\right)} (y - K\langle x,y\rangle_{\mathcal{L}} x)$ |
| $\Gamma_{x \to y}(v)$ | $\frac{\lambda_x^K}{\lambda_y^K} \mathrm{gyr}[y, -x]v$ | $v - \frac{K\langle y,v\rangle_{\mathcal{L}}}{1+K\langle x,y\rangle_{\mathcal{L}}}(x+y)$ |
| $\mathrm{Exp}_x(v)$ | $x \oplus_M \left( \tanh\left( \sqrt{|K|}\frac{\lambda_x^K\|v\|}{2} \right) \frac{v}{\sqrt{|K|}\|v\|} \right)$ | $\cosh\left( \sqrt{|K|}\|v\|_{\mathcal{L}} \right) x + \sinh\left( \sqrt{|K|}\|v\|_{\mathcal{L}} \right) \frac{v}{\sqrt{|K|}\|v\|_{\mathcal{L}}}$ |
| References | (Ganea et al., 2018) (Skopek et al., 2020; Ungar, 2022a) | (Petersen, 2006; Skopek et al., 2020) |

Table 12: Riemannian operators on the Beltrami–Klein model ($K < 0$).

| Operators | $\mathbb{K}^n_K = \left\{ x \in \mathbb{R}^n \mid \|x\|^2 < -\frac{1}{K} \right\}$ |
|---|---|
| $g_x(v,w)$ | $\frac{\langle v,w \rangle}{1+K\|x\|^2} - \frac{K\langle x,v\rangle\langle x,w\rangle}{\left(1+K\|x\|^2\right)^2}$ |
| $\mathrm{d}(x,y)$ | $\frac{2}{\sqrt{-K}} \tanh^{-1}\left( \sqrt{-K} \frac{\| -x \oplus_E y\|}{1+\sqrt{1+K\|-x\oplus_E y\|^2}} \right)$ |
| $\mathrm{Exp}_x(v)$ | $x \oplus_E \mathrm{Exp}_0\left( \frac{1}{\sqrt{1+K\|x\|^2}}v - \frac{K\langle x,v\rangle}{(1+\sqrt{1+K\|x\|^2})(1+K\|x\|^2)} x \right)$ |
| $\mathrm{Log}_x(y)$ | $\frac{1}{\lambda_{\tilde{x}}^K} \left( \pi_{\mathbb{P}^n_K \to \mathbb{K}^n_K} \right)_{*,\tilde{x}}(\mathrm{Log}_0(-x \oplus_E y)), \quad \tilde{x} = \pi_{\mathbb{K}^n_K \to \mathbb{P}^n_K}(x)$ |
| References | (Ungar, 2022b; Chen et al., 2025b) |

and $r \in \mathbb{R}$, the gyro operations are defined as

$$\text{Möbius addition}: x \oplus_M y = \frac{\left(1 - 2K\langle x,y\rangle - K\|y\|^2\right)x + \left(1 + K\|x\|^2\right)y}{1 - 2K\langle x,y\rangle + K^2\|x\|^2\|y\|^2}, \quad (15)$$

$$\text{Möbius scalar multiplication}: r \otimes_M x = \frac{\tanh\left(r \tanh^{-1}\left(\sqrt{-K}\|x\|\right)\right)}{\sqrt{-K}} \frac{x}{\|x\|}, \quad (16)$$

$$\text{Einstein addition}: x \oplus_E y = \frac{1}{1 - K\langle x,y\rangle}\left( x + \frac{1}{\gamma_x}y - K\frac{\gamma_x}{1+\gamma_x}\langle x,y\rangle x \right), \quad (17)$$

$$\text{Einstein scalar multiplication}: r \otimes_E x = \frac{\tanh\left(r \tanh^{-1}\left(\sqrt{-K}\|x\|\right)\right)}{\sqrt{-K}} \frac{x}{\|x\|}. \quad (18)$$

where $\gamma_x = 1/\sqrt{1+K\|x\|^2}$ is called the gamma factor. Interestingly, the scalar gyromultiplications are identical under the Möbius and Einstein gyrovector spaces.

The Poincaré ball and hyperboloid admit closed-form Riemannian operators, as summarized in Tab. 11. The parallel transport over the Poincaré ball requires the notion of gyration (Ungar, 2022b):

$$\mathrm{gyr}[x,y]z = \ominus_M (x \oplus_M y) \oplus_M (x \oplus_M (y \oplus_M z)), \forall x,y,z \in \mathbb{P}^n_K. \quad (19)$$

Chen et al. (2025a, Sec. 5.6) studied the Riemannian structure over the Beltrami–Klein ball. The Beltrami–Klein ball is isometric to the Poincaré ball by

$$\pi_{\mathbb{K}^n_K \to \mathbb{P}^n_K} : x \in \mathbb{K}^n_K \longmapsto \frac{1}{1 + \sqrt{1 + K\|x\|^2}}x \in \mathbb{P}^n_K, \quad (20)$$

$$\pi_{\mathbb{P}^n_K \to \mathbb{K}^n_K} : x \in \mathbb{P}^n_K \longmapsto \frac{2}{1 - K\|x\|^2}x \in \mathbb{K}^n_K. \quad (21)$$

By the above isometries, Chen et al. (2025a, Sec. 5.6) introduced the closed-form expression for the Riemannian operators on the Beltrami–Klein ball, as summarized in Tab. 12. Particularly, the Riemannian exponential and logarithmic maps at the zero vector $\mathbf{0}$ are identical under the Beltrami–Klein and Poincaré ball models:

$$\text{Exp}_{\mathbf{0}}(v) = \tanh(\sqrt{|K|}\|v\|)\frac{v}{\sqrt{|K|}\|v\|}, \quad \forall v \in T_{\mathbf{0}}\mathcal{H}, \tag{22}$$

$$\text{Log}_{\mathbf{0}}(x) = \tanh^{-1}(\sqrt{|K|}\|x\|)\frac{x}{\sqrt{|K|}\|x\|}, \quad \forall x \in \mathcal{H}, \tag{23}$$

with $\mathcal{H} \in \{\mathbb{K}^n_K, \mathbb{P}^n_K\}$.

As shown by Chen et al. (2025b)[Sec. 5.4 and 5.6], both the Möbius and Einstein gyrovector operations can be expressed by their Riemannian geometries

$$x \oplus_{\mathcal{H}} y = \text{Exp}_x\left(\Gamma_{\mathbf{0} \to x}(\text{Log}_{\mathbf{0}}(y))\right), \tag{24}$$

$$t \otimes_{\mathcal{H}} x = \text{Exp}_{\mathbf{0}}(t\,\text{Log}_{\mathbf{0}}(x)), \tag{25}$$

where $\oplus_{\mathcal{H}}$ and $\otimes_{\mathcal{H}}$ are the gyroaddition and gyromultiplication under the corresponding model.

## D.2 Geometries of the SPD manifold

Tabs. 13 and 14 summarizes the associated Riemannian operators and properties. Following Tab. 10, we further make the following notation. Given any SPD points $P, Q \in \mathcal{S}^n_{++}$ and tangent vectors $V, W \in T_P\mathcal{S}^n_{++}$, we denote $\widetilde{V} = \text{Chol}_{*,P}(V)$, $\widetilde{W} = \text{Chol}_{*,P}(W)$, $L = \text{Chol}\,P$, and $K = \text{Chol}\,Q$. The corresponding diagonal matrix with their diagonal elements are denoted as $\widetilde{\mathbb{V}}, \widetilde{\mathbb{W}}, \mathbb{L}$, and $\mathbb{K}$, respectively. For the parallel transport under the BWM, we only present the case where $P, Q$ are commuting matrices, $i.e. P = U\Sigma U^\top$ and $Q = U\Delta U^\top$.

The $\text{O}(n)$-invariant Euclidean metric on $\mathcal{S}^n$ (Thanwerdas and Pennec, 2023) is

$$\langle V, W \rangle^{(\alpha,\beta)} = \alpha\langle V, W \rangle + \beta\,\text{tr}(V)\,\text{tr}(W), \quad \text{with } \min(\alpha, \alpha + n\beta) > 0. \tag{26}$$

*Remark* D.1. We make the following remarks w.r.t. the geometries on the SPD manifold.

- **PEM & EM.** When the power equals 1, the associated PEM is reduced to the Euclidean Metric (EM) (Thanwerdas and Pennec, 2023, Sec. 3.1).
- **Incompleteness & Riemannian exponentiation.** As PEM and BWM are incomplete, their Riemannian exponential maps are locally defined. As shown by Malagò et al. (2018, Prop. 9) and implied by Chen et al. (2024c); Thanwerdas and Pennec (2023), the restricted domains are

$$\begin{aligned} \text{PEM: } & P^\theta + P_{\theta*,P}(V) \in \mathcal{S}^n_{++}, \\ \text{BWM: } & \mathcal{L}_P[V] + I \in \mathcal{S}^n_{++}. \end{aligned} \tag{27}$$

The above restriction can be solved numerically, such as ReEig (Huang et al., 2017):

$$\widetilde{S} = U\max(\epsilon I, \Sigma)U^\top, \tag{28}$$

where $S \overset{\text{Eig}}{:=} U\Sigma U^\top$ is the Eigendecomposition.

## D.3 Geometries of the Grassmannian

As the set of linear subspaces, the Grassmannian can naturally be represented by any of the orthonormal bases, which is called the OrthoNormal Basis (ONB) perspective. Under this perspective, the Grassmannian is the quotient of the Stiefel manifold (Bendokat et al., 2024), denoted as $\text{Gr}(p, n) \cong \text{St}(p, n)/\text{O}(p)$. Each point is an equivalence class:

$$\text{Gr}(p, n) = \{[U] : [U] := \{\widetilde{U} \in \text{St}(p, n) \mid \widetilde{U} = UR, R \in \text{O}(p)\}\}. \tag{29}$$

By abuse of notation, we use $[U]$ and $U$ interchangeably for elements of $\text{Gr}(p, n)$. Each tangent space can be identified as a subspace of a corresponding tangent space on the Stiefel manifold, which is called horizontal space. Therefore, every tangent vector can be identified with a tangent

Table 13: The Riemannian operators under LEM, AIM, and PEM on the SPD manifold.

| Operators | LEM | AIM | PEM |
|---|---|---|---|
| $g_P(V,W)$ | $\langle \log_{*,P}(V), \log_{*,P}(W) \rangle^{(\alpha,\beta)}$ | $\langle P^{-1}V, WP^{-1} \rangle^{(\alpha,\beta)}$ | $\frac{1}{\theta^2}\langle \mathrm{P}_{\theta*,P}(V), \mathrm{P}_{\theta*,P}(W) \rangle^{(\alpha,\beta)}$ |
| $\mathrm{Log}_P Q$ | $(\log_{*,P})^{-1}\left[\log(Q)-\log(P)\right]$ | $P^{\frac{1}{2}}\log\left(P^{-\frac{1}{2}}QP^{-\frac{1}{2}}\right)P^{\frac{1}{2}}$ | $(\mathrm{P}_{\theta*,P})^{-1}\left(Q^\theta - P^\theta\right)$ |
| $\Gamma_{P\to Q}(V)$ | $(\log_{*,Q})^{-1}\circ\log_{*,P}(V)$ | $(QP^{-1})^{\frac{1}{2}}V(P^{-1}Q)^{\frac{1}{2}}$ | $(\mathrm{P}_{\theta*,Q})^{-1}\circ\mathrm{P}_{\theta*,P}(V)$ |
| $\mathrm{Exp}_P(V)$ | $\exp\left(\log(P)+\log_{*,P}(V)\right)$ | $P^{\frac{1}{2}}\exp\left(P^{-\frac{1}{2}}VP^{-\frac{1}{2}}\right)P^{\frac{1}{2}}$ | $\left(P^\theta + \mathrm{P}_{\theta*,P}(V)\right)^{\frac{1}{\theta}}$ |
| Invariance | Lie group bi-invariance O($n$)-invariance | Lie group left-invariance GL($n$)-invariance | O($n$)-invariance |
| References | (Arsigny et al., 2005) (Thanwerdas and Pennec, 2023) | (Pennec et al., 2006) (Thanwerdas and Pennec, 2019) | (Dryden et al., 2010) (Thanwerdas and Pennec, 2023) (Chen et al., 2024c) |

Table 14: The Riemannian operators under BWM and LCM on the SPD manifold.

| Operators | LCM | BWM |
|---|---|---|
| $g_P(V,W)$ | $\langle \lfloor\widetilde{V}\rfloor, \lfloor\widetilde{W}\rfloor \rangle + \langle \widetilde{\mathbb{V}}\widetilde{\mathbb{L}}^{-1}, \widetilde{\mathbb{W}}\widetilde{\mathbb{L}}^{-1} \rangle$ | $\frac{1}{2}\langle \mathcal{L}_P[V], W \rangle$ |
| $\mathrm{Log}_P Q$ | $(\mathrm{Chol}^{-1})_{*,L}\left[\lfloor K\rfloor - \lfloor L\rfloor + \mathbb{L}\,\mathrm{Dlog}(\mathbb{L}^{-1}\mathbb{K})\right]$ | $(PQ)^{\frac{1}{2}}+(QP)^{\frac{1}{2}}-2P$ |
| $\Gamma_{P\to Q}(V)$ | $(\mathrm{Chol}^{-1})_{*,K}\left[\lfloor\widetilde{V}\rfloor + \mathbb{K}\mathbb{L}^{-1}\widetilde{\mathbb{V}}\right]$ | $U\left[\sqrt{\frac{\delta_i+\delta_j}{\sigma_i+\sigma_j}}\left[U^\top V U\right]_{ij}\right]U^\top$ |
| $\mathrm{Exp}_P(V)$ | $\mathrm{Chol}^{-1}\left[\lfloor L\rfloor + \lfloor\widetilde{V}\rfloor + \mathbb{L}\,\mathrm{Dexp}(\mathbb{L}^{-1}\widetilde{V})\right]$ | $P+V+\mathcal{L}_P[V]P\mathcal{L}_P[V]$ |
| Invariance | Lie group bi-invariance | O($n$)-invariance |
| References | (Lin, 2019) | (Bhatia et al., 2019) (Thanwerdas and Pennec, 2023) |

vector in the horizontal space, called horizontal lift[2]. Under this identification, each tangent vector $V \in T_P\mathrm{Gr}(p,n)$ can be represented as

$$V = P_\perp B, \text{ with } B \in \mathbb{R}^{(n-p)\times p}, \tag{30}$$

where $P_\perp \in \mathrm{St}(n-p,n)$ is the orthogonal complement of $P$.

Another perspective is called the Projector Perspective (PP). As shown by Bendokat et al. (2024), the Grassmannian is an embedded submanifold of $\mathcal{S}^n$:

$$\widetilde{\mathrm{Gr}}(p,n) = \{P \in \mathcal{S}^n : P^2 = P, \mathrm{rank}(P) = p\}. \tag{31}$$

Therefore, each point can be represented as an $n \times n$ symmetric matrix. Under this perspective, any tangent vector $V \in T_P\widetilde{\mathrm{Gr}}(p,n)$ at $P \in \widetilde{\mathrm{Gr}}(p,n)$ can be represented as

$$V = Q\begin{pmatrix} 0 & B^T \\ B & 0 \end{pmatrix}Q^T, \text{ with } B \in \mathbb{R}^{(n-p)\times p}, \tag{32}$$

where $Q\widetilde{I}_{p,n}Q^\top = P$.

Supposing $P$ and $Q$ are the points on the Grassmannian $\mathrm{Gr}(p,n)$ ($\widetilde{\mathrm{Gr}}(p,n)$), and $V$ and $W$ are the tangent vectors over $T_P\mathrm{Gr}(p,n)$ ($T_P\widetilde{\mathrm{Gr}}(p,n)$), Tab. 15 summarizes the associated Riemannian operators following the notation in Tab. 10.

*Remark* D.2. We make the following remarks w.r.t. the Riemannian operators over the Grassmannian.

- **Cut locus & logarithm.** The Grassmannian Riemannian logarithm does not exists for any pair of $P$ and $Q$. As shown by Bendokat et al. (2024, Sec. 5), $\mathrm{Log}_P(Q)$ exists only if $P$ and $Q$ are not in each other's cut locus. However, this can be numerically solved, such as Bendokat et al. (2024, Alg. 5.3) or using Moore–Penrose inverse for the inverse in the ONB logarithm (Nguyen, 2022).

---

[2]In this paper, the tangent vector under the ONB perspective is always considered as the horizontal lift.

Table 15: Riemannian operators on the Grassmannian.

| Operators | $\mathrm{Gr}(p,n)$ | $\widetilde{\mathrm{Gr}}(p,n)$ |
|---|---|---|
| $g_P(V,W)$ | $\langle V,W \rangle$ | $\frac{1}{2}\langle V,W \rangle$ |
| $\mathrm{Log}_P Q$ | $O \arctan(\Sigma) R^\top$ $(I_n - PP^\top)Q(P^\top Q)^{-1} \overset{\mathrm{SVD}}{:=} O\Sigma R^\top$ | $\frac{1}{2}[\log\left((I_n - 2Q)(I_n - 2P)\right), P]$ |
| $\Gamma_{P \to Q}(V)$ | $\left(\begin{pmatrix} PR & O \end{pmatrix} \begin{pmatrix} -\sin(\Sigma) \\ \cos(\Sigma) \end{pmatrix} O^T + (I - OO^T)\right)V$ $\mathrm{Log}_P(Q) \overset{\mathrm{SVD}}{:=} O\Sigma R^\top$ | $\exp([\log_P(Q), P])V \exp(-[\log_P(Q), P])$ |
| $\mathrm{Exp}_P V$ | $\begin{pmatrix} PR & O \end{pmatrix} \begin{pmatrix} \cos(\Sigma) \\ \sin(\Sigma) \end{pmatrix} R^\top$ $V \overset{\mathrm{SVD}}{:=} O\Sigma R^\top$ | $\exp([V,P])P\exp(-[V,P])$ |
| References | (Edelman et al., 1998) (Bendokat et al., 2024) | (Batzies et al., 2015) (Bendokat et al., 2024) |

- **PP & ONB logarithm.** The matrix logarithm shown in the PP logarithm does not support backpropagation, as it can not be calculated by the SVD like the SPD matrix. However, the PP logarithm can be calculated via the ONB logarithm (Nguyen et al., 2024, Prop. 3.12). The latter can be backpropagated by the SVD. In this way, the PP logarithm can be integrated into the Pytorch deep learning framework.

# E DISCUSSIONS ON THE RIEMANNIAN FC AND CONVOLUTIONAL LAYER

## E.1 ADDITIONAL DISCUSSIONS ON THE ORTHOGONAL BASIS

When the inner product $g_E$ on $T_E\mathcal{M}$ is the standard inner product, we use familiar $\{e_i\}_{i=1}^m$ the orthonormal basis. However, when $g_E$ is not standard, $\{e_i\}_{i=1}^m$ might not be orthonormal. In this case, we can always find one associated to $\{e_i\}_{i=1}^m$ by a linear isometry. We rewrite the inner product $g_E$ as

$$g_E(V,W) = \langle f(V), f(W) \rangle = f(V)^\top f(W), \forall V, W \in T_E\mathcal{M} \cong \mathbb{R}^m, \tag{33}$$

where $f$ is the linear isometry that pulls back the standard inner product $\langle \cdot, \cdot \rangle$ to $g_E$. Then, $\{B_i\}_{i=1}^m = \{f^{-1}(e_i)\}_{i=1}^m$ is the standard orthonormal bases over $\{T_E\mathcal{M}, g_E\}$.

## E.2 RIEMANNIAN FULLY CONNECTED LAYERS UNDER ISOMETRIC GEOMETRY

As isometric Riemannian metrics commonly arise in various geometries (Thanwerdas and Pennec, 2022; Chen et al., 2024d; Bendokat et al., 2024), we discuss the construction of Riemannian FC layers under isometries. The following theorem demonstrates that a Riemannian FC layer under isometric metrics can be computed by the following procedure: mapping, applying the Riemannian FC layer, and remapping. This result will be applied in our concrete examples of the SPD and Grassmannian FC layers.

We denote FC transformation as $Y = \mathcal{F}(X; \mathbf{A}, \mathbf{P})$, with $\mathbf{P} = \{P_i \in \mathcal{N}\}_{i=1}^m$ and $\mathbf{A} = \{A_i \in T_{P_i}\mathcal{N}\}_{i=1}^m$ as the FC parameters.

**Theorem E.1** (Isometric FC Layers). *Given $n$-dimensional Riemannian manifolds $\left\{\widetilde{\mathcal{N}}, g^{\widetilde{\mathcal{N}}}\right\}$ and $\left\{\mathcal{N}, g^{\mathcal{N}}\right\}$ with a Riemannian isometry $\phi^{\mathcal{N}} : \widetilde{\mathcal{N}} \to \mathcal{N}$, and $m$-dimensional Riemannian manifolds $\left\{\widetilde{\mathcal{M}}, g^{\widetilde{\mathcal{M}}}\right\}$ and $\left\{\mathcal{M}, g^{\mathcal{M}}\right\}$ with $\phi^{\mathcal{M}} : \widetilde{\mathcal{M}} \to \mathcal{M}$ as a Riemannian isometry mapping origin $E^{\widetilde{\mathcal{M}}} \in \widetilde{\mathcal{M}}$ into the origin $E \in \mathcal{M}$, the Riemannian FC layer $\widetilde{\mathcal{F}} : \widetilde{\mathcal{N}} \to \widetilde{\mathcal{M}}$ can be calculated by $\mathcal{F} : \mathcal{N} \to \mathcal{M}$:*

$$\widetilde{\mathcal{F}}\left(\widetilde{X}; \widetilde{\mathbf{P}}, \widetilde{\mathbf{A}}\right) = \left(\phi^{\mathcal{M}}\right)^{-1}\left(\mathcal{F}\left(\phi^{\mathcal{N}}(\widetilde{X}); \mathbf{P}, \mathbf{A}\right)\right), \tag{34}$$

*where $\widetilde{\mathbf{P}} = \left\{\widetilde{P}_i \in \widetilde{\mathcal{N}}\right\}_{i=1}^m$ and $\widetilde{\mathbf{A}} = \left\{\widetilde{A}_i \in T_{\widetilde{P}_i}\widetilde{\mathcal{N}}\right\}_{i=1}^m$ are the FC parameters of $\widetilde{\mathcal{F}}$, while $\mathbf{P} = \left\{\phi^{\mathcal{N}}(\widetilde{P}_i)\right\}_{i=1}^m$ and $\mathbf{A} = \left\{\phi^{\mathcal{N}}_{*,\widetilde{P}_i}(\widetilde{A}_i)\right\}_{i=1}^m$ are the FC parameters of $\mathcal{F}$.*

*Proof.* First we show the correspondence between the standard orthonormal bases $\{\widetilde{B}_i \in \widetilde{\mathcal{M}}\}$ and $\{B_i \in \mathcal{M}\}$. Obviously, $\{\widetilde{B}_i \in \widetilde{\mathcal{M}}\}$ is orthonormal iff $\{B_i \in \mathcal{M}\}$ is orthonormal. We only need to show the standardness. The Riemannian metric $g^{\widetilde{\mathcal{M}}}$ has the following:

$$
\begin{aligned}
g_{\widetilde{E}}^{\widetilde{\mathcal{M}}}(V, W) &\stackrel{(1)}{=} g_E^{\mathcal{M}}\left(\phi_{*,\widetilde{E}}^{\mathcal{M}}(V), \phi_{*,\widetilde{E}}^{\mathcal{M}}(V)\right) \\
&= \left\langle f \circ \phi_{*,\widetilde{E}}^{\mathcal{M}}(V), f \circ \phi_{*,\widetilde{E}}^{\mathcal{M}}(V) \right\rangle,
\end{aligned}
\tag{35}
$$

where $f$ is the linear isomorphism that pulls back the standard Frobenius inner product to $g_E^{\mathcal{M}}$. Here, (1) comes from the isometry. Therefore, for each $i$, we have the following

$$
\begin{aligned}
\widetilde{B}_i &= (f \circ \phi_{*,\widetilde{E}}^{\mathcal{M}})^{-1}(E_i) \\
&\stackrel{(1)}{=} \left(\phi_{*,\widetilde{E}}^{\mathcal{M}}\right)^{-1}(B_i),
\end{aligned}
\tag{36}
$$

where (1) comes from $B_i = f^{-1}(E_i), \forall i = 1, \cdots, n$.

We now demonstrate the correspondence between the FC layers as follows:

$$
\begin{aligned}
Y &= \operatorname{Exp}_{\widetilde{E}}^{\widetilde{\mathcal{M}}}\left(\sum_{i=1}^m \left(\langle \operatorname{Log}_{\widetilde{P}_i}^{\widetilde{\mathcal{N}}}(\widetilde{X}), \widetilde{A}_i \rangle_{\widetilde{P}_i}^{\widetilde{\mathcal{N}}} \widetilde{B}_i\right)\right) \\
&\stackrel{(1)}{=} (\phi^{\mathcal{M}})^{-1}\left(\operatorname{Exp}_E^{\mathcal{M}}\left(\phi_{*,\widetilde{E}}^{\mathcal{M}}\left[\sum_{i=1}^m \left(\langle \operatorname{Log}_{P_i}^{\mathcal{N}}(X), A_i \rangle_{P_i}^{\mathcal{N}} \widetilde{B}_i\right)\right]\right)\right) \\
&\stackrel{(2)}{=} (\phi^{\mathcal{M}})^{-1}\left(\operatorname{Exp}_E^{\mathcal{M}}\left(\sum_{i=1}^m \left(\langle \operatorname{Log}_{P_i}^{\mathcal{N}}(X), A_i \rangle_{P_i}^{\mathcal{N}} B_i\right)\right)\right),
\end{aligned}
\tag{37}
$$

where $B_i = \phi_{*,\widetilde{E}}^{\mathcal{M}}(\widetilde{B}_i)$, $A_i = \phi_{*,\widetilde{P}_i}^{\mathcal{N}}(\widetilde{A}_i)$, $X = \phi^{\mathcal{N}}(\widetilde{X})$, and $P_i = \phi^{\mathcal{N}}(\widetilde{P}_i)$. The above derivation comes from the following.

(1) The isometry of $\phi^{\mathcal{M}}$ and $\phi^{\mathcal{N}}$;
(2) The linearity of $\phi_{*,\widetilde{E}}^{\mathcal{M}}$.

$\square$

### E.3  RIEMANNIAN FULLY CONNECTED LAYERS UNDER PRODUCT GEOMETRY

Now, we discuss Thm. 3.3 under product geometry

**Theorem E.2.** *Following the notation in Thm. 3.3, the Riemannian FC layer $\mathcal{F}(\cdot) : (\mathcal{N})^c \to \mathcal{M}$ for the input $(X_1 \in \mathcal{N}, \cdots, X_c \in \mathcal{N}) = X \in (\mathcal{N})^c$ is*

$$
Y = \operatorname{Exp}_E^{\mathcal{M}}\left(\sum_{i=1}^m \sum_{j=1}^c \langle \operatorname{Log}_{P_{ij}}^{\mathcal{N}}(X), A_{ij} \rangle_{P_{ij}}^{\mathcal{N}} B_i\right),
\tag{38}
$$

*where $P_{ij} \in \mathcal{N}$ and $A_{ij} \in T_{P_{ij}}\mathcal{N}$ are the FC parameters.*

*Proof.* By product geometry, we have

$$
(\mathcal{N})^c \ni P_i = (P_{i1} \in \mathcal{N}, \cdots, P_{ic} \in \mathcal{N}),
\tag{39}
$$

$$
T_{P_i}(\mathcal{N})^c \ni A_i = (A_{i1} \in T_{P_{i1}}\mathcal{N}, \cdots, A_{ic} \in T_{P_{ic}}\mathcal{N}).
\tag{40}
$$

The above implies that

$$
\langle \operatorname{Log}_{P_i}^{(\mathcal{N})^c}(X), A_i \rangle_{P_i}^{(\mathcal{N})^c} = \sum_{j=1}^c \langle \operatorname{Log}_{P_{ij}}^{\mathcal{N}}(X), A_{ij} \rangle_{P_{ij}}^{\mathcal{N}}.
\tag{41}
$$

$\square$

### E.4 RIEMANNIAN FULLY CONNECTED LAYERS AND MANIFOLD EMBEDDING

In several applications (Chami et al., 2019; López et al., 2021; Zhao et al., 2023; Nguyen et al., 2024), embedding Euclidean features into non-Euclidean manifolds often yields superior results. A common approach can be expressed as $\mathrm{Exp}_E(Ax + b)$, which maps Euclidean features to the tangent space at the origin via a linear layer, followed by applying the exponential map at the origin. This method has been adopted in various embeddings, including hyperbolic (Chami et al., 2019; Fu et al., 2024), SPD (Zhao et al., 2023), and Grassmannian spaces (Nguyen et al., 2024, Sec. 3.4.2). Our framework offers a novel intrinsic interpretation, showing that this operation respects the Riemannian FC layer between the Euclidean space and the target manifold.

**Proposition E.3.** *The Riemannian FC layer from a standard Euclidean space $\mathbb{R}^n$ to an $m$-dimensional target manifold $\mathcal{M}$, namely $\mathcal{F}(\cdot): \mathbb{R}^n \to \mathcal{M}$, is given by*

$$\mathcal{F}(x) = \mathrm{Exp}_E(Ax + b), \tag{42}$$

*where $A \in \mathbb{R}^{n \times m}$ and $b \in \mathbb{R}^m$ are the transformation matrix and biasing vector, respectively.*

*Proof.* By Thm. 3.3, we have the following

$$
\begin{aligned}
Y &\overset{(1)}{=} \mathrm{Exp}_E^{\mathcal{M}}\left(\sum_{i=1}^m \left(\langle \mathrm{Log}_{p_i}^{\mathrm{Euc}}(x), a_i \rangle_{p_i}^{\mathrm{Euc}} B_i\right)\right), \\
&\overset{(2)}{=} \mathrm{Exp}_E^{\mathcal{M}}\left(\sum_{i=1}^m \left(\langle x - p_i, a_i \rangle B_i\right)\right), \\
&\overset{(3)}{=} \mathrm{Exp}_E^{\mathcal{M}}\left(\sum_{i=1}^m \left(\langle x - p_i, a_i \rangle f^{-1}(e_i)\right)\right), \\
&\overset{(4)}{=} \mathrm{Exp}_E^{\mathcal{M}}\left(f^{-1}\left(\sum_{i=1}^m \langle x - p_i, a_i \rangle e_i\right)\right), \\
&\overset{(5)}{=} \mathrm{Exp}_E^{\mathcal{M}}\left(f^{-1}\left(\bar{A}x + \bar{b}\right)\right), \\
&\overset{(6)}{=} \mathrm{Exp}_E^{\mathcal{M}}\left(Ax + b\right).
\end{aligned}
\tag{43}
$$

The above comes from the following,

(1) $p_i, a_i \in \mathbb{R}^n$, and $\{B_i\}$ are the orthonormal bases over $\{T_E \mathcal{M}, g_E\}$;
(2) The Euclidean logarithm and metric become the familiar vector operation:

$$\mathrm{Log}_{p_i}^{\mathrm{Euc}}(x) = x - p_i$$
$$\langle v, w \rangle_p^{\mathrm{Euc}} = \langle v, w \rangle, \forall p \in \mathbb{R}^n, \forall v, w \in T_p \mathbb{R}^n;$$

(3) $f$ is the linear isomorphism pulling the standard inner product back to $g_E$; $\{e_i\}$ are the standard orthonormal bases over the standard inner product;
(4) Linearity of $f^{-1}$;
(5) $\sum_{i=1}^m \langle x - p_i, a_i \rangle e_i$ has the form of affine transformation;
(6) As $f^{-1}$ has matrix representation, $f^{-1}(x) = \tilde{A}x$, we have

$$
\begin{aligned}
f^{-1}\left(\bar{A}x + \bar{b}\right) &= \tilde{A}\left(\bar{A}x + \bar{b}\right) \\
&= \tilde{A}\bar{A}x + \tilde{A}\bar{b}.
\end{aligned}
\tag{44}
$$

Setting $A = \tilde{A}\bar{A}$ and $b = \tilde{A}\bar{b}$, one can obtain the result.

$\square$

### E.5 RELATION WITH THE CONVOLUTION IN MANIFOLDNET

Chakraborty et al. (2020) also proposed a convolution operation for manifolds. However, as their formulation is based on the weighted Fréchet mean, it is unable to alter the manifold dimension, such as dimensionality reduction. In contrast, our framework allows for modifications in both the channel and manifold dimensions, providing greater flexibility.

## F  COMPARISON OF OUR HYPERBOLIC FC LAYERS AGAINST PREVIOUS HYPERBOLIC LINEAR LAYERS

Tab. 16 extends Tab. 1, comparing our hyperbolic FC layers against previous hyperbolic linear layers.

Table 16: Comparison of hyperbolic linear layers. Here, we consider the transformation from an $n$-dimensional hyperbolic space to an $m$-dimensional one.

| Method | Model | Mechanism | Formulation | Parameters | References |
|--------|-------|-----------|-------------|------------|------------|
| Möbius | $\mathbb{P}_K^n$ | Tangent | $\mathrm{Exp}_\mathbf{0}(M\,\mathrm{Log}_\mathbf{0}(x))$ | $M \in \mathbb{R}^{m \times n}$ | (Ganea et al., 2018, Def. 3.2) |
| Klein | $\mathbb{K}_K^n$ | Tangent | $\mathrm{Exp}_\mathbf{0}(M\,\mathrm{Log}_\mathbf{0}(x))$ | $M \in \mathbb{R}^{m \times n}$ | (Mao et al., 2024, Thm. 9) |
| Lorentz | $\mathbb{H}_K^n$ | Spacetime | $\begin{bmatrix} \frac{\sqrt{\|Wx\|^2 - 1/K}}{v^\top x} v^\top \\ W \end{bmatrix} x$ | $M \in \mathbb{R}^{m \times (n+1)}$ $v \in \mathbb{R}^{n+1}$ | (Chen et al., 2022, Sec 3.1) |
| Poincaré FC | $\mathbb{P}_K^n$ | Poincaré | $w\left(1 + \sqrt{1 - K\|w\|^2}\right)^{-1}$ $w = \left((-K)^{-\frac{1}{2}} \sinh\left(\sqrt{-K}\, v_k(x)\right)\right)_{k=1}^m$ $v_k$ is defined by Shimizu et al. (2021, Eq. (6)) | $\{z_i \in \mathbb{R}^n\}_{i=1}^m$ $\{\gamma_i \in \mathbb{R}\}_{i=1}^m$ | (Shimizu et al., 2021, Sec. 3.2) |
| Ours | $\mathbb{P}_K^n, \mathbb{K}_K^n, \mathbb{H}_K^n$ | Riemannian | Thms. 4.1 and 4.2 | $\{z_i \in \mathbb{R}^n\}_{i=1}^m$ $\{\gamma_i \in \mathbb{R}\}_{i=1}^m$ | Thms. 4.1 and 4.2 |

## G  ADDITIONAL DETAILS ON THE SPD FULLY CONNECTED LAYERS

### G.1  RELATION WITH THE GYRO SPD FULLY CONNECTED LAYERS

This subsection demonstrates that our SPD FC layers subsume three gyro SPD FC layers under LEM, AIM, and LCM. This follows directly from Prop. 3.2, as one can readily verify that the point-to-hyperplane distance we used is identical to the corresponding gyro distances under these three metrics. To clarify this relationship more clearly, we compare the final expressions.

We first review some related SPD gyro structures (Nguyen and Yang, 2023). Given $P, Q$ in $\{\mathcal{S}_{++}^n, g\}$ with $g$ as AIM, LEM or LCM, and $t \in \mathbb{R}$, the gyro structures induced by $g$ are defined as follows:

$$\text{Gyro addition: } P \oplus Q = \mathrm{Exp}_P\left(\Gamma_{I \to P}\left(\mathrm{Log}_I(Q)\right)\right), \tag{45}$$

$$\text{Scalar gyromultiplication: } t \otimes P = \mathrm{Exp}_I\left(t\,\mathrm{Log}_I(P)\right), \tag{46}$$

$$\text{Gyro inverse: } \ominus P = -1 \otimes P = \mathrm{Exp}_I\left(-\mathrm{Log}_I(P)\right), \tag{47}$$

$$\text{Gyro inner product: } \langle P, Q \rangle_{\mathrm{gr}} = \langle \mathrm{Log}_I(P), \mathrm{Log}_I(Q) \rangle_I, \tag{48}$$

where $\mathrm{Log}_I$ and $\langle \cdot, \cdot \rangle_I$ is the Riemannian logarithm and metric at the identity matrix $I$. As shown by Nguyen (2022), the gyro addition and scalar product under AIM, LEM, and LCM form gyrovector spaces.

Based on these gyro structures, Nguyen et al. (2024) introduced the gyro SPD FC layers under AIM, LEM, and LCM, respectively. We review their results in the following.

**Theorem G.1** (Gyro SPD FC Layers (Nguyen et al., 2024))**.** *The gyro SPD FC layers under standard LEM, AIM, and LCM are*

$$LEM : Y = \exp\left(V^{\mathrm{LE}}\right), V_{ij}^{\mathrm{LE}} = \begin{cases} v_{ii}^{\mathrm{LE}}(S), & \text{if } i = j \\ \frac{1}{\sqrt{2}} v_{ij}^{\mathrm{LE}}(S), & \text{if } i > j \\ V_{ji}^{\mathrm{LE}}, & \text{otherwise} \end{cases} \tag{49}$$

$$AIM : Y = \exp\left(V^{\mathrm{AI}}\right), V_{ij}^{\mathrm{AI}} = \begin{cases} v_{ii}^{\mathrm{AI}}(S) + \eta \sum_{k=1}^m v_{kk}^{\mathrm{AI}}(S), & \text{if } i = j \\ \frac{1}{\sqrt{2}} v_{ij}^{\mathrm{AI}}(S), & \text{if } i > j \\ V_{ji}^{\mathrm{AI}}, & \text{otherwise} \end{cases} \tag{50}$$

$$LCM : Y = V^{\mathrm{LC}}(V^{\mathrm{LC}})^\top, V_{ij}^{\mathrm{LC}} = \begin{cases} \exp\left(v_{ii}^{\mathrm{LC}}(S)\right), & \text{if } i = j \\ v_{ij}^{\mathrm{LC}}(S), & \text{if } i > j \\ 0, & \text{otherwise} \end{cases} \tag{51}$$

*where* $\eta = \frac{1}{n}\left(\frac{1}{\sqrt{1+n\beta}} - 1\right)$, *and* $v_{ij}^g = \langle \ominus P_{ij} \oplus S, W_{ij} \rangle_{\mathrm{gr}}$ *with $g$ as LEM, AIM, or LCM. Here,* $P_{ij}, W_{ij} \in \mathcal{S}_{++}^n, \forall i \geq j, i, j = 1, \cdots, m$.

**Proposition G.2.** *Our LEM $((\alpha, \beta) = (1, 0))$, AIM $((\alpha, \beta) = (1, \beta))$, and LCM SPD FC layers incorporate the LEM, AIM, and LCM gyro SPD FC layers, respectively.*

*Proof.* Comparing Thm. G.1 with our Thm. 4.3, we only need to show the equality of $v_{ij}$ in the gyro and our framework:

$$v_{ij}^g \stackrel{(1)}{=} \left\langle \mathrm{Log}_{P_{ij}}(S), \Gamma_{I \to P_{ij}}(\mathrm{Log}_I(W_{ij})) \right\rangle_{P_{ij}}, \tag{52}$$

where (1) has been proved in Prop. 3.2. Setting $A_{ij} = \Gamma_{I \to P}(\mathrm{Log}_I(W_{ij})) \in T_{P_{ij}}\mathcal{S}_{++}^n$, we recover Eqs. (94), (95) and (97) for each metric. $\square$

## G.2 RELATION WITH THE FLAT SPD FULLY CONNECTED LAYERS

Nguyen et al. (2025) proposed two SPD FC layers based on flat LEM and LCM. However, as shown by Nguyen et al. (2025, App. B. 2.2), it has the same formulation as the LEM and LCM gyro SPD FC layer, respectively.

## G.3 TRIVIALIZED SPD FULLY CONNECTED LAYERS

**Theorem G.3** (Trivialized SPD FC Layers). *Trivializing each $P_{ij}$ in Thm. 4.3 as $\mathrm{Exp}_I(\gamma_{ij}[Z_{ij}])$, $v_{ij}(S)$ under different metrics can be further simplified:*

$$LEM : \langle \log(S), Z_{ij} \rangle^{(\alpha, \beta)} - \gamma_{ij} \|Z_{ij}\|^{(\alpha, \beta)}, \tag{53}$$

$$AIM : \left\langle \log\left(\exp\left(-\frac{\gamma_{ij}}{2}[Z_{ij}]\right) S \exp\left(-\frac{\gamma_{ij}}{2}[Z_{ij}]\right)\right), Z_{ij} \right\rangle^{(\alpha, \beta)}, \tag{54}$$

$$PEM : \left\langle S^\theta - (I + \theta\gamma_{ij}[Z_{ij}]), Z_{ij} \right\rangle^{(\alpha, \beta)}, \tag{55}$$

$$LCM : \left\langle \lfloor K \rfloor + \mathrm{Dlog}(\mathbb{K}) - \left(\gamma_{ij}\lfloor [Z_{ij}] \rfloor + \frac{1}{2}\gamma_{ij}\mathbb{D}([Z_{ij}])\right), \lfloor Z_{ij} \rfloor + \frac{1}{2}\mathbb{Z}_{ij} \right\rangle, \tag{56}$$

*where $\|\cdot\|^{(\alpha, \beta)}$ is the norm induced by $\langle \cdot, \cdot \rangle^{(\alpha, \beta)}$, and $\mathbb{D}(\cdot)$ returns a diagonal matrix with diagonal elements from the input square matrix.*

*Proof.* **LEM:**

$$\begin{aligned}
\langle \log(S) - \log(P_{ij}), Z_{ij} \rangle^{(\alpha, \beta)} &\stackrel{(1)}{=} \langle \log(S) - \gamma_{ij}[Z_{ij}], Z_{ij} \rangle^{(\alpha, \beta)} \\
&\stackrel{(2)}{=} \langle \log(S), Z_{ij} \rangle^{(\alpha, \beta)} - \gamma_{ij} \|Z_{ij}\|^{(\alpha, \beta)},
\end{aligned} \tag{57}$$

The above comes from the following.

(1) Eq. (110);
(2) $[Z_{ij}] = \frac{Z_{ij}}{\|Z_{ij}\|^{(\alpha, \beta)}}$.

**AIM:** This can be obtained by the following:

$$\exp\left(\gamma_{ij}[Z_{ij}]\right)^{-\frac{1}{2}} = \exp\left(-\frac{\gamma_{ij}}{2}[Z_{ij}]\right). \tag{58}$$

**PEM:** This can be obtained by Eq. (111).

**LCM:**

$$\begin{aligned}
&\left\langle \lfloor K \rfloor - \lfloor L_{ij} \rfloor + \mathrm{Dlog}(\mathbb{K}\mathbb{L}_{ij}^{-1}), \lfloor Z_{ij} \rfloor + \frac{1}{2}\mathbb{Z}_{ij} \right\rangle \\
&= \left\langle \lfloor K \rfloor + \mathrm{Dlog}(\mathbb{K}) - (\lfloor L_{ij} \rfloor + \mathrm{Dlog}(\mathbb{L}_{ij})), \lfloor Z_{ij} \rfloor + \frac{1}{2}\mathbb{Z}_{ij} \right\rangle \\
&\stackrel{(1)}{=} \left\langle \lfloor K \rfloor + \mathrm{Dlog}(\mathbb{K}) - \left(\gamma_{ij}\lfloor [Z_{ij}] \rfloor + \frac{1}{2}\gamma_{ij}\mathbb{D}([Z_{ij}])\right), \lfloor Z_{ij} \rfloor + \frac{1}{2}\mathbb{Z}_{ij} \right\rangle,
\end{aligned} \tag{59}$$

where (2) comes from Eq. (112). $\square$

*Remark* G.4. Due to the incompleteness of PEM and BWM, their exponential maps at $I$, $\mathrm{Exp}_I(V)$, are well-defined locally:

$$\text{PEM: } I + \theta V \in \mathcal{S}_{++}^n,$$
$$\text{BWM: } I + \frac{1}{2}V \in \mathcal{S}_{++}^n. \tag{60}$$

The above restriction can be solved numerically, such as ReEig (Huang et al., 2017):

$$\widetilde{S} = U \max(\epsilon I, \Sigma) U^\top, \tag{61}$$

where $S \overset{\mathrm{Eig}}{:=} U \Sigma U^\top$ is the eigendecomposition.

### G.4 TRIVIALIZED SPD MULTINOMIAL LOGISTIC REGRESSION

In our implementation, we trivialize the SPD parameters in the SPD MLR as Sec. 3.3. The SPD MLRs proposed by Chen et al. (2024c) under five geometries can be further simplified. For simplicity, we do not involve the power deformation (Chen et al., 2024c).

**Theorem G.5** (Trivialized SPD MLRs). [↓] *Given $C$ classes and an SPD feature $S$, the SPD MLRs, $p(y = k \mid S \in \mathcal{S}_{++}^n)$, are proportional to*

$$\textit{LEM} : \exp\left[ \langle \log(S), Z_k \rangle^{(\alpha,\beta)} - \gamma_k \left\| Z_k \right\|^{(\alpha,\beta)} \right], \tag{62}$$

$$\textit{AIM} : \left[ \exp \left\langle \log\left( \exp\left(-\frac{\gamma_k}{2}[Z_k]\right) S \exp\left(-\frac{\gamma_k}{2}[Z_k]\right) \right), Z_k \right\rangle^{(\alpha,\beta)} \right], \tag{63}$$

$$\textit{PEM} : \frac{1}{\theta} \exp\left[ \left\langle S^\theta - (I + \theta \gamma_k[Z_k]), Z_k \right\rangle^{(\alpha,\beta)} \right], \tag{64}$$

$$\textit{LCM} : \exp\left[ \left\langle \lfloor K \rfloor + \mathrm{Dlog}(\mathbb{K}) - \left( \gamma_k \lfloor [Z_k] \rfloor + \frac{1}{2}\gamma_k \mathbb{D}([Z_k]) \right), \lfloor Z_k \rfloor + \frac{1}{2}\mathbb{Z}_k \right\rangle \right], \tag{65}$$

$$\textit{BWM:} \exp\left[ \frac{1}{2} \left\langle (P_k S)^{\frac{1}{2}} + (S P_k)^{\frac{1}{2}} - 2P_k, \mathcal{L}_{P_k}(L_k Z_k L_k^\top) \right\rangle \right], \tag{66}$$

*where $Z_k \in T_I \mathcal{S}_{++}^n \backslash \{0\}$ is a symmetric matrix, $L_k = \mathrm{Chol}(P_k)$ is the Cholesky factor of $P_k$ with $P_k = (I + \frac{1}{2}\gamma_k[Z_k])^2$. Here $\{Z_k \in \mathcal{S}^n\}_{k=1}^C$ and $\{\gamma_k \in \mathbb{R}\}_{k=1}^C$ are the MLR parameters.*

*Proof.* For each class $k$, the expression of $v_k$ in the SPD MLR (Chen et al., 2024c, Thm. 4.2) has been reviewed in Sec. J.5. For MLR under each metric $g$, we parameterize the each parameter $P_k \in \mathcal{S}_{++}^n$ by $Z_k$ and $\gamma_k$ by

$$P_k = \mathrm{Exp}_I^g(\gamma_k[Z_k]), \tag{67}$$

with $[Z_k]$ as the unit vector of $Z_k$. Under this parameterization, the MLRs under LEM, AIM, PEM, and LCM can be further simplified, which has been implied by Thm. G.3. □

*Remark* G.6. Similar to the SPD FC layer, due to the incompleteness of PEM and BWM, the associated parameterization should follow

$$\text{PEM: } I + \theta \gamma_k[Z_k] \in \mathcal{S}_{++}^n, \tag{68}$$
$$\text{BWM: } I + \frac{1}{2}\gamma_k[Z_k] \in \mathcal{S}_{++}^n. \tag{69}$$

## H REVIEW OF PREVIOUS GRASSMANNIAN TRANSFORMATION LAYERS

This section briefly reviews several popular Grassmannian transformation layers.

**FRMap + ReOrth.** Given input Grassmannian $X \in \mathrm{Gr}(p, q)$, Huang et al. (2018) first used Full Rank Map (FRMap) to first transform the input orthonormal matrices of subspaces to new matrices by a linear mapping function, and then applied QR decomposition to recover the orthogonality:

$$Y = \mathcal{Q}(WX), \tag{70}$$

where $W \in \mathbb{R}^{m \times n}$ is a row-wisely orthogonal parameter, and $\mathcal{Q}(\cdot)$ returns the orthogonal matrix in the QR decomposition.

**PP & ONB Scaling.** Nguyen (2022); Nguyen and Yang (2023) proposed matrix scaling for the PP and ONB Grassmannian, respectively. Given $P = XX^\top \in \widetilde{\mathrm{Gr}}(p,n)$ with $X \in \mathrm{Gr}(p,n)$, the operations are defined as

$$\textbf{PP: } Y = \exp\left(\begin{bmatrix} 0 & W * B \\ -(W * B)^T & 0 \end{bmatrix}\right) \widetilde{I}_{p,n} \exp\left(-\begin{bmatrix} 0 & W * B \\ -(W * B)^T & 0 \end{bmatrix}\right), \quad (71)$$

$$\textbf{ONB: } Y = \exp\left(\begin{bmatrix} 0 & W * B \\ -(W * B)^T & 0 \end{bmatrix}\right) I_{p,n}, \tag{72}$$

where $*$ denotes the Hadamard product and $B \in \mathbb{R}^{(n-p) \times p}$ is a Euclidean parameter. Here, $X = \exp\left(\begin{bmatrix} 0 & B \\ -B^T & 0 \end{bmatrix}\right) I_{p,n}$.

**GrTrans.** Nguyen and Yang (2023) adopted the Grassmannian Gyro group translation (GrTrans) to transform the ONB and PP Grassmannian features. Given $X \in \widetilde{\mathrm{Gr}}(p,n)$ ( or $X \in \mathrm{Gr}(p,n)$), the operation is defined as

$$Y = W \oplus X, \tag{73}$$

where $\oplus$ is the Grassmannian PP (ONB) gyro addition (Nguyen and Yang, 2023, Sec. 2.3), and $W \in \widetilde{\mathrm{Gr}}(p,n)$ (or $W \in \mathrm{Gr}(p,n)$) is a Grassmannian parameter.

# I    ADDITIONAL EXPERIMENTAL DETAILS AND RESULTS

## I.1    HYPERBOLIC SPACES

### I.1.1    DATASETS

**Disease (Anderson and May, 1991).** It represents a disease propagation tree, simulating the SIR disease transmission model, with each node representing either an infection or a non-infection state.

**Airport (Zhang and Chen, 2018).** It is a transductive dataset where nodes represent airports and edges represent the airline routes as from OpenFlights.org.

**Pubmed (Namata et al., 2012).** This is a standard benchmark describing citation networks where nodes represent scientific papers in the area of medicine, edges are citations between them, and node labels are academic (sub)areas.

**Cora (Sen et al., 2008).** It is a citation network where nodes represent scientific papers in the area of machine learning, edges are citations between them, and node labels are academic (sub)areas.

### I.1.2    IMPLEMENTATION DETAILS

We follow the official implementations of HNN[3] (Ganea et al., 2018), HNN++[4] (Shimizu et al., 2021) and HyboNet [5] (Chen et al., 2022) to conduct the experiments. For the Einstein transformation in the Beltrami–Klein model, we carefully implement it according to the original paper (Mao et al., 2024). We adopt the settings as HGCN[6] (Chami et al., 2019) for the link prediction task.

**Details on main experiments.** Following the HNN implementation (Ganea et al., 2018; Chami et al., 2019; Mao et al., 2024), the baseline encoder consists of two transformation layers: the first maps the input feature dimension to 16, and the second maps 16 to 16. The transformation layers could be our HFC layers or others like Möbius, Einstein, Poincaré FC, LorentzTan, or Lorentz linear layer. Each transformation is followed by an activation layer $\mathrm{Exp}_o(\mathrm{ReLu}(\mathrm{Log}_o(x)))$, where $o$ is the origin in each model. Following HNN, we also adopt the bias translation after each HFC layer, *i.e.*, $x \oplus b = \mathrm{Exp}_x(\Gamma_{o \to x} \mathrm{Log}_o(x))$. We use the Adam optimizer (Kingma, 2015) with a learning rate of $1e^{-2}$. We fine-tune the dropout of transformation weight and weight decay.

---

[3]https://github.com/dalab/hyperbolic_nn

[4]https://github.com/mil-tokyo/hyperbolic_nn_plusplus

[5]https://github.com/chenweize1998/fully-hyperbolic-nn

[6]https://github.com/HazyResearch/hgcn

**Details on ablations on the RResNet.** We employ a hyperbolic transformation layer to map each input vector into an 8-dimensional vector in the Poincaré ball. The network consists of two residual blocks, each configured with different hidden dimensions and varying numbers of horospheres. We use the Adam optimizer (Kingma, 2015) and fine-tune hyperparameters, such as the learning rate and weight decay.

## I.2 SPD MANIFOLDS

### I.2.1 DATASETS

**Radar[7] (Brooks et al., 2019).** It consists of 3,000 synthetic radar signals equally distributed in 3 classes.

**HDM05[8] (Müller et al., 2007).** It consists of 2,343 skeleton-based motion capture sequences executed by different actors. Each frame consists of 3D coordinates of 31 joints. We remove the under-represented clips, trimming the dataset down to 2,326 instances scattered throughout 122 classes. We randomly select 50% of the samples from each category for training and the remaining 50% for testing.

**FPHA[9] (Garcia-Hernando et al., 2018).** It includes 1,175 skeleton-based first-person hand gesture videos of 45 different categories with 600 clips for training and 575 for testing. Each frame contains the 3D coordinates of 21 hand joints.

For the HDM05 and FPHA datasets, we preprocess each sequence using the code[10] provided by Vemulapalli et al. (2014) to normalize body part lengths and ensure invariance to scale and view.

### I.2.2 SPD MODELING

For our SPDNNs, we follow Wang et al. (2024a); Nguyen et al. (2024) to model each sample into a multi-channel SPD tensor. For the Radar dataset, we follow Wang et al. (2024a) to use the temporal convolution followed by a covariance pooling layer to obtain a multi-channel covariance $[c, 20, 20]$ tensor. For the HDM05 and FPHA datasets, we follow Nguyen et al. (2024, Sec. D.2.2) to model each skeleton sequence into a multi-channel covariance tensor $[c, n, n]$. Specifically, we first identify the closest left (right) neighbor of every joint based on their distance to the hip (wrist) joint, and then combine the 3D coordinates of each joint and those of its left (right) neighbor to create a feature vector for the joint. For a given frame $t$, we compute its Gaussian embedding (Lovrić et al., 2000):

$$Y_t = (\det \Sigma_t)^{-\frac{1}{n+1}} \left[ \begin{array}{cc} \Sigma_t + \mu_t (\mu_t)^T & \mu_t \\ (\mu_t)^T & 1 \end{array} \right], \tag{74}$$

where $\mu_t$ and $\Sigma_t$ are the mean vector and covariance matrix computed from the set of feature vectors within the frame. The lower part of matrix $\log (Y_t)$ is flattened to obtain a vector $\tilde{v}_t$. All vectors $\tilde{v}_t$ within a time window $[t, t+c-1]$, where $c$ is determined from a temporal pyramid representation of the sequence (the number of temporal pyramids is set to 2 in our experiments), are used to compute a covariance matrix as

$$Z_t = \frac{1}{c} \sum_{i=t}^{t+c-1} (\tilde{v}_i - \overline{v}_t) (\tilde{v}_i - \overline{v}_t)^T, \tag{75}$$

where $\overline{v}_t = \frac{1}{c} \sum_{i=t}^{t+c-1} \tilde{v}_i$. The resulting $\{Z_t\}$ is the input covariance tensor. On the FPHA dataset, we generate the covariance based on three sets of neighbors: left, right, and vertical (bottom) neighbors.

For GyroLE, GyroAI, GyroLC, and GyroSPD++, the input are similar to our SPDNNs. For other SPD baselines, such as SPDNet, SPDNetBN, LieBN, MLR, and RResNet, each sequence is represented by a global covariance representation (Huang and Van Gool, 2017; Brooks et al., 2019). The sizes of the covariance matrices are $20 \times 20$, $93 \times 93$, and $63 \times 63$ for Radar, HDM05, and FPHA datasets, respectively.

---

[7] https://www.dropbox.com/s/dfnlx2bnyh3kjwy/data.zip?dl=0

[8] https://resources.mpi-inf.mpg.de/HDM05/

[9] https://github.com/guiggh/hand_pose_action

[10] https://ravitejav.weebly.com/kbac.html

Table 17: Training hyer-parameters in SPDNNs

| Dataset | Model | $\theta$ | Optimizer | Learning Rate |
|---------|-------|----------|-----------|---------------|
| Radar | SPDNN-LEM | N/A | AMSGrad | $5e^{-3}$ |
| | SPDNN-AIM | 0.25 | AMSGrad | $5e^{-4}$ |
| | SPDNN-PEM | N/A | AMSGrad | $1e^{-2}$ |
| | SPDNN-LCM | 0.25 | AMSGrad | $5e^{-4}$ |
| | SPDNN-BWM | N/A | AMSGrad | $5e^{-4}$ |
| HDM05 | SPDNN-LEM | N/A | SGD | $5e^{-3}$ |
| | SPDNN-AIM | N/A | SGD | $5e^{-3}$ |
| | SPDNN-PEM | N/A | AMSGrad | $1e^{-3}$ |
| | SPDNN-LCM | N/A | AMSGrad | $1e^{-3}$ |
| | SPDNN-BWM | N/A | AMSGrad | $1e^{-3}$ |
| FPHA | SPDNN-LEM | N/A | AMSGrad | $1e^{-4}$ |
| | SPDNN-AIM | N/A | AMSGrad | $1e^{-4}$ |
| | SPDNN-PEM | N/A | AMSGrad | $1e^{-3}$ |
| | SPDNN-LCM | -0.25 | AMSGrad | $1e^{-3}$ |
| | SPDNN-BWM | -0.25 | AMSGrad | $1e^{-4}$ |
| NTU60 | SPDNN-LEM | N/A | SGD | $1e^{-3}$ |
| | SPDNN-AIM | N/A | AMSGrad | $1e^{-4}$ |
| | SPDNN-PEM | N/A | AMSGrad | $5e^{-4}$ |
| | SPDNN-LCM | 0.25 | AMSGrad | $5e^{-4}$ |
| | SPDNN-BWM | 0.25 | AMSGrad | $1e^{-3}$ |

### I.2.3 IMPLEMENTATION DETAILS

**Comparative methods.** We follow the official Pytorch code of SPDNetBN[11] to implement SPDNet and SPDNetBN. For LieBN[12], we focus on the instantiation under AIM and LCM, while for RRes-Net[13], we implement the ones induced by LEM and AIM. For SPD MLR[14], we implement the ones induced by LCM. For GyroLE, GyroAI, GyroLC, and GyroSPD++, we re-implemented them based on the original paper (Nguyen and Yang, 2023; Nguyen et al., 2024).

**SPDNNs.** On all datasets, we employ a single convolutional kernel for global convolution, *i.e.*, applying a global receptive field across the channel dimension. The output dimensions of the SPD convolutional layer are $8 \times 8$, $34 \times 34$, $22 \times 22$, and $11 \times 11$ for the Radar, HDM05, FPHA, and NTU60 datasets, respectively. We primarily use the AMSGrad (Reddi et al., 2018) optimizer, except for SPDNN-LEM and SPDNN-AIM on the HDM05 dataset and SPDNN-LEM on the NTU60, where SGD (Robbins and Monro, 1951) is employed. Weight decay is set to zero, except for SPDNN-PEM on the FPHA dataset, where it is $5e^{-4}$. The matrix power in SPDNN-PEM is set as 0.5 for the Radar, and 0.25 for the other three datasets. Since matrix power can deform the latent Riemannian metric (Chen et al., 2024c, Fig. 1), we also apply matrix power $(\cdot)^{\theta}$ before the convolutional layer in SPDNN-AIM, -LCM, and -BWM to activate the latent geometries. The batch size is set to 30 with a training epoch of 150 with early stopping. Tab. 17 summarizes the training hyper-parameters.

### I.2.4 TRAINING EFFICIENCY

Tab. 18 presents the average training time per epoch of each SPD network. We have the following observations:

- **The efficiency of SPDNN varies across metrics.** The most efficient metric is LCM, where our model even achieves comparable efficiency to the vanilla SPDNet. However, AIM and

---

[11] https://proceedings.neurips.cc/paper_files/paper/2019/file/
6e69ebbfad976d4637bb4b39de261bf7-Supplemental.zip

[12] https://github.com/GitZH-Chen/LieBN

[13] https://github.com/CUAI/Riemannian-Residual-Neural-Networks

[14] https://github.com/GitZH-Chen/SPDMLR

Table 18: Training efficiency (second / epoch).

| Method | Geometrtry | Radar | HDM05 | FPHA | NTU60 |
|---|---|---|---|---|---|
| SPDNet | N/A | 0.66 | 0.50 | 0.28 | 3.08 |
| SPDNetBN | AIM | 1.25 | 0.94 | 0.58 | 6.14 |
| SPDResNet-AIM | AIM | 0.96 | 1.23 | 0.69 | 6.84 |
| SPDResNet-LEM | LEM | 0.77 | 0.55 | 0.30 | 3.17 |
| SPDNetLieBN-AIM | AIM | 1.21 | 1.15 | 0.97 | 8.85 |
| SPDNetLieBN-LCM | LCM | 1.10 | 1.11 | 0.59 | 5.96 |
| SPDNetMLR | LCM | 0.66 | 5.46 | 0.88 | 4.94 |
| GyroLE | LEM | 0.79 | 2.86 | 1.59 | 10.57 |
| GyroLC | LCM | 0.66 | 1.49 | 0.78 | 5.99 |
| GyroAI | AIM | 0.99 | 22.80 | 12.62 | 26.76 |
| GyroSPD++-AIM | AIM | 5.09 | 103.57 | 66.35 | 125.05 |
| GyroSPD++-LEM | LEM | 0.99 | 0.95 | 0.66 | 7.58 |
| GyroSPD++-LCM | LCM | 0.66 | 0.70 | 0.37 | 5.74 |
| SPDNN-LEM | LEM | 0.86 | 0.74 | 0.63 | 5.79 |
| SPDNN-AIM | AIM | 4.84 | 101.80 | 65.42 | 124.41 |
| SPDNN-PEM | PEM | 1.09 | 7.10 | 1.57 | 8.71 |
| SPDNN-LCM | LCM | 0.65 | 0.59 | 0.35 | 3.72 |
| SPDNN-BWM | BWM | 6.07 | 110.51 | 71.67 | 139.48 |

BWM demonstrate significant computational burden, primarily due to their complex Riemannian computations.

- **Our trivialization improves efficiency.** Compared with the LCM-based SPDNetMLR, SPDNN-LCM achieves much lower training time. This improvement can be partially attributed to our trivialization, which simplifies the final expression of MLR (Sec. G.4) and eliminates the need for computationally expensive Riemannian optimization. Besides, SPDNN consistently outperforms GyroSPD++ under LEM, LCM, and AIM in terms of efficiency. This advantage arises because our trivialization not only simplifies the expression of the FC and MLR layers, but also reduces the number of parameters.

## I.3 GRASSMANNIAN MANIFOLDS

**Grassmannian modeling.** As Grassmannian descriptors can be derived by the SVD of the covariance (Huang et al., 2018; Nguyen and Yang, 2023), we map the multi-channel Radar covariance into a $[c, n, p]$ ONB Grassmannian tensor via the SVD decomposition. The PP Grassmannian features can be derived from the ONB Grassmannian features via the isometry $\pi(\cdot) : \mathrm{Gr}(p, n) \to \widetilde{\mathrm{Gr}}(p, n)$:

$$\pi(U) = UU^\top, \forall U \in \mathrm{Gr}(p, n). \tag{76}$$

**Implementation details.** Since GrNet (Huang et al., 2018) is officially implemented by Matlab, we carefully re-implemented it using PyTorch. Additionally, as both GryroGr and GryroGr-Scaling do not release official code, we re-implemented them based on the original papers (Nguyen and Yang, 2023). For all comparative methods, we use SGD with a learning rate of $5e^{-2}$. For training our ONB and PP GrNNs, we use AMSGrad with a learning rate of $5e^{-3}$. The batch size is set to 30 with a training epoch of 150.

## I.4 HARDWARE

On the HDM05 and FPHA datasets, SPDNet, RResNet, SPDNetBN, SPDNetLieBN, and MLR require SVD operations on relatively large matrices, which are more efficiently executed on a CPU. As a result, these methods are implemented on a CPU, whereas all other cases are executed on a single A6000 GPU.

# J PROOFS

## J.1 PROOF OF PROP. 3.2

*Proof.* **Euclidean spaces.** We first review the following facts about Euclidean space:

- the origin is the zero vector $\mathbf{0}$;
- the standard orthonormal basis over $T_0 \mathbb{R}^m \cong \mathbb{R}^m$ is $\{e_i\}_{i=1}^m$;
- $\mathrm{Log}_p(x) = x - p$ and $\langle \cdot, \cdot \rangle_p = \langle \cdot, \cdot \rangle$ for any $x, p \in \mathbb{R}^n$;
- point-to-hyperplane distance is $d(x, H_{a,p}) = \frac{|\langle x-p,a \rangle|}{\|a\|}$.

Putting the above together, one can recover Eq. (1).

**Poincaré balls.** This exactly corresponds to the derivation of the Poincaré FC layer (Shimizu et al., 2021, App. D.3).

**SPD gyrovector spaces.** Nguyen et al. (2024) proposed three gyro SPD FC layers based on the gyrovector structures under LEM, LCM, and AIM, respectively. The below discussion summarizes the proof in Nguyen et al. (2024, Apps. J-L).

In the SPD gyro FC layer, the origin of the SPD manifold is the identity matrix $I$. Given a metric among LEM, LCM, and AIM, let $\{B_i\}_{i=1}^d$ be an orthonormal basis over $T_I \mathcal{S}_{++}^m$, where $d = {}^{n(n+1)}/_2$ is the dimension of $\mathcal{S}_{++}^n$. The SPD gyro FC layer is defined by solving the following equations

$$\mathrm{sign}\left(\langle \mathrm{Log}_I(Y), B_k \rangle_I\right) \mathrm{d}(Y, H_{B_k,I}) = \langle W_k, \ominus P_k \oplus X \rangle_{gr}, \quad 1 \le k \le m, \tag{77}$$

where $\ominus$ and $\oplus$ are gyro operations (Nguyen et al., 2024, Apps. G.2-G.4) and $\langle \cdot, \cdot \rangle_{gr}$ is the gyro inner product (Nguyen et al., 2024, App. G.7). Here, each $W_k \in \mathcal{S}_{++}^n$ and $P_k \in \mathcal{S}_{++}^n$ are FC parameters. By Prop. 3.2 in Nguyen et al. (2024), the RHS of Eq. (77) is equal to $\langle \mathrm{Log}_{P_k}(X), \Gamma_{I \to P_k}(\mathrm{Log}_I(W_k)) \rangle_{P_k}$. Setting $A_k = \Gamma_{I \to P_k}(\mathrm{Log}_I(W_k)) \in T_{P_k} \mathcal{S}_{++}^n$, one can recover Eq. (3). $\qquad\square$

## J.2 PROOF OF THM. 3.3

*Proof.* The Riemannian signed distance from a point $Y \in \mathcal{M}$ to a Riemannian hyperplane over $\mathcal{M}$ is

$$\bar{\mathrm{d}}(Y, \widetilde{H}_{A,P}) = \frac{\langle \mathrm{Log}_P^{\mathcal{M}} Y, A \rangle_P^{\mathcal{M}}}{\|A\|_P^{\mathcal{M}}}, \tag{78}$$

where $\widetilde{H}_{A,P}$ is a Riemannian hyperplane parameterized by $P \in \mathcal{M}$ and $A \in T_P \mathcal{M}$. Therefore, the signed distance from $Y$ to $\widetilde{H}_{B_i,E}$ is

$$\widetilde{\mathrm{d}}(Y, \widetilde{H}_{B_i,E}) = \frac{\langle \mathrm{Log}_E^{\mathcal{M}}(Y), B_i \rangle_E^{\mathcal{M}}}{\|B_i\|_E^{\mathcal{M}}}$$

$$\overset{(1)}{=} \langle \mathrm{Log}_E^{\mathcal{M}}(Y), B_i \rangle_E^{\mathcal{M}} \tag{79}$$

where (1) comes from the orthonormality of $B_i$.

Setting Eq. (79) equal to $v_i(X)$, we have

$$\langle \mathrm{Log}_E^{\mathcal{M}}(Y), B_i \rangle_E^{\mathcal{M}} = \langle \mathrm{Log}_{P_i}^{\mathcal{N}}(X), A_i \rangle_{P_i}^{\mathcal{N}}. \tag{80}$$

The above equation indicates

$$\mathrm{Log}_E^{\mathcal{M}}(Y) = \sum_{i=1}^m \left( \langle \mathrm{Log}_{P_i}^{\mathcal{N}}(X), A_i \rangle_{P_i}^{\mathcal{N}} B_i \right). \tag{81}$$

$\qquad\square$

## J.3 PROOF OF THM. 4.1

To simplify the Riemannian FC layer with the gyro structure, we first prove a useful lemma.

**Lemma J.1.** *We assume that the manifold $\mathcal{M}$ admits a gyrogroup (Nguyen, 2022, Def. 2.2) defined by*[15]

$$x \oplus y = \mathrm{Exp}_x \left( \Gamma_{e \to x} \left( \mathrm{Log}_e (y) \right) \right), \forall p, q \in \mathcal{M}, \tag{82}$$

*where $e \in \mathcal{M}$ is the origin of the manifold. Then, we have the following*

$$\left\langle \mathrm{Log}_p(x), a \right\rangle_p = \left\langle \mathrm{Log}_e(\ominus p \oplus x), \Gamma_{p \to e}(a) \right\rangle_e, \quad \forall x, p \in \mathcal{M} \text{ and } \forall a \in T_p \mathcal{M}. \tag{83}$$

*Proof.* **Credit of the proof:** Eq. (82) comes from Nguyen and Yang (2023, Eq. (1)), who demonstrated that several geometries admit gyrogroups based on this definition. The prototype of Eq. (83) comes from App. I by Nguyen et al. (2024), which only deals with SPD matrices. Here, we further extend the result into general gyrogroups.

Denoting $\ominus p$ as the gyro inverse of $p$ ($\ominus p \oplus p = e$), we have

$$x \stackrel{(1)}{=} p \oplus (\ominus p \oplus x) \stackrel{(2)}{=} \mathrm{Exp}_p \left( \Gamma_{e \to p} \left( \mathrm{Log}_e \left( \ominus p \oplus x \right) \right) \right)$$
$$\stackrel{(3)}{\Rightarrow} \mathrm{Log}_p(x) = \Gamma_{e \to p} \left( \mathrm{Log}_e \left( \ominus p \oplus x \right) \right). \tag{84}$$

The above comes from the following,

(1) Left cancellation law of the gyrogroup (Ungar, 2022a, Thms. 1.13).
(2) Definition of gyro addition.
(3) Applying both sides with $\mathrm{Log}_p(\cdot)$.

By the last equation, we have

$$\left\langle \mathrm{Log}_p(x), a \right\rangle_p = \left\langle \Gamma_{e \to p} \left( \mathrm{Log}_e \left( \ominus p \oplus x \right) \right), a \right\rangle_p$$
$$\stackrel{(1)}{=} \left\langle \mathrm{Log}_e \left( \ominus p \oplus x \right), \Gamma_{p \to e}(a) \right\rangle_e, \tag{85}$$

where (1) comes from

- Parallel transport preserving the norm (Do Carmo and Flaherty Francis, 1992, Sec. 3.1)
- $\Gamma_{p \to e} \circ \Gamma_{e \to p}(v) = v, \forall v \in T_e \mathcal{M}$.

$\square$

Now we begin to prove Thm. 4.1.

*Proof of Thm. 4.1.* In both geometries, the origins are defined as the zero vector, as it is the identity element in their own gyrovector spaces. We first deal with the Poincaré ball followed by the Beltrami–Klein model.

**Poincaré ball:** The Riemannian metric at the identity element is

$$\left\langle v, w \right\rangle_{\mathbf{0}} = 4 \left\langle v, w \right\rangle, \forall v, w \in T_{\mathbf{0}} \mathbb{P}_K^m. \tag{86}$$

Obviously, $\{ \frac{1}{4} e_i \}_{i=1}^m$ is an orthonormal basis. Lem. J.1 implies

$$\left\langle \mathrm{Log}_{p_i}(x), a_i \right\rangle_{p_i} \frac{1}{4} e_i \stackrel{(1)}{=} \left\langle \mathrm{Log}_{\mathbf{0}}(-p_i \oplus_{\mathrm{M}} x), \Gamma_{p_i \to \mathbf{0}}(a_i) \right\rangle_{\mathbf{0}} \frac{1}{4} e_i$$
$$\stackrel{(2)}{=} \left\langle \mathrm{Log}_{\mathbf{0}}(-p_i \oplus_{\mathrm{M}} x), \Gamma_{p_i \to \mathbf{0}}(a_i) \right\rangle e_i \tag{87}$$
$$\stackrel{(3)}{=} \left\langle \mathrm{Log}_{\mathbf{0}}(-p_i \oplus_{\mathrm{M}} x), z_i \right\rangle e_i.$$

The above comes from the following.

(1) Lem. J.1 and $\ominus_{\mathrm{M}} p = -p, \quad \forall p \in \mathbb{P}_K^n$.
(2) Eq. (86).
(3) $a_i = \Gamma_{\mathbf{0} \to p_i}(z_i)$.

---

[15]We assume all the involved Riemannian operators are well-defined.

**Beltrami–Klein model:** The Riemannian metric at the identity element is

$$\langle v, w \rangle_{\mathbf{0}} = \langle v, w \rangle, \forall v, w \in T_{\mathbf{0}} \mathbb{K}_K^m. \tag{88}$$

Obviously, $\{e_i\}_{i=1}^m$ is an orthonormal basis. Lem. J.1 and Eq. (24) implies that the above reasoning for the Poincaré ball can be transferred into the Beltrami–Klein model:

$$
\begin{aligned}
\left\langle \mathrm{Log}_{p_i}(x), a_i \right\rangle_{p_i} e_i &\overset{(1)}{=} \left\langle \mathrm{Log}_{\mathbf{0}}(-p_i \oplus_{\mathrm{E}} x), \Gamma_{p_i \to \mathbf{0}}(a_i) \right\rangle_{\mathbf{0}} e_i \\
&= \left\langle \mathrm{Log}_{\mathbf{0}}(-p_i \oplus_{\mathrm{E}} x), \Gamma_{p_i \to \mathbf{0}}(a_i) \right\rangle e_i \\
&\overset{(2)}{=} \left\langle \mathrm{Log}_{\mathbf{0}}(-p_i \oplus_{\mathrm{E}} x), z_i \right\rangle e_i,
\end{aligned}
\tag{89}
$$

The above comes from the following.

(1) Lem. J.1, Eq. (24), and $\ominus_{\mathrm{E}} p = -p$, .
(2) $a_i = \Gamma_{\mathbf{0} \to p_i}(z_i)$.

$\square$

### J.4 PROOF OF THM. 4.2

*Proof.* We only need to show the origin, the tangent space at the origin, and the inner product and an orthonormal basis over the tangent space at the origin.

The hyperboloid is isometric to the Poincaré ball by the following diffeomorphism (Lee, 2006):

$$\pi_{\mathbb{P}_K^n \to \mathbb{H}_K^n}(x) = \left( \frac{1}{\sqrt{|K|}} \frac{1 - K\|x\|^2}{1 + K\|x\|^2}; \frac{2x^T}{1 + K\|x\|^2} \right)^{\top}. \tag{90}$$

The origin of hyperboloid is therefore defined as

$$e := \pi_{\mathbb{P}_K^n \to \mathbb{H}_K^n}(\mathbf{0}) = \left( \frac{1}{\sqrt{|K|}}, 0 \cdots, 0 \right)^{\top}. \tag{91}$$

The Riemannian metric and tangent space at $e$ are

$$T_e \mathbb{H}_K^n = \{(0, v^{\top})^{\top} | v \in \mathbb{R}^n\}, \tag{92}$$

$$\langle (0, v^{\top})^{\top}, (0, w^{\top})^{\top} \rangle_e = \langle v, w \rangle, \quad \forall (0, v^{\top})^{\top}, (0, w^{\top})^{\top} \in T_e \mathbb{H}_K^n. \tag{93}$$

Therefore, $\{(0, e_i^{\top})^{\top}\}_{i=1}^m$ is an orthonormal basis of $T_e \mathbb{H}_K^n$ with $e_i \in \mathbb{R}^n$.

Putting the above with Tab. 11, we can manifest Thm. 3.3 in the hyperboloid geometry. $\square$

### J.5 PROOF OF THM. 4.3

*Proof.* In the following proof, we first present the expressions of several operators under different metrics, including $v_{ij}(S)$, standard orthonormal bases, and Riemannian exponentiation at the origin. Then, we begin to prove the theorem. In this proof, we follow all the notation as the theorem.

$v_{ij}(S)$ **under different metrics:** The expressions are implied by Chen et al. (2024c, Thm. 4.2):

$$\mathrm{LEM} : \langle \log(S) - \log(P_{ij}), Z_{ij} \rangle^{(\alpha,\beta)}, \tag{94}$$

$$\mathrm{AIM} : \left\langle \log(P_{ij}^{-\frac{1}{2}} S P_{ij}^{-\frac{1}{2}}), Z_{ij} \right\rangle^{(\alpha,\beta)}, \tag{95}$$

$$\mathrm{PEM} : \frac{1}{\theta} \left\langle S^{\theta} - P_{ij}^{\theta}, Z_{ij} \right\rangle^{(\alpha,\beta)}, \tag{96}$$

$$\mathrm{LCM} : \left\langle \lfloor K \rfloor - \lfloor L_{ij} \rfloor + \mathrm{Dlog}(\mathbb{K} \mathbb{L}_{ij}^{-1}), \lfloor Z_{ij} \rfloor + \frac{1}{2} \mathbb{Z}_{ij} \right\rangle, \tag{97}$$

$$\mathrm{BWM} : \frac{1}{2} \left\langle (P_{ij}S)^{\frac{1}{2}} + (SP_{ij})^{\frac{1}{2}} - 2P_{ij}, \mathcal{L}_{P_{ij}}(L_{ij} Z_{ij} L_{ij}^{\top}) \right\rangle. \tag{98}$$

**Standard orthonormal bases:** Next, we show the standard orthonormal bases over $T_I \mathcal{S}_{++}^n$ under different metrics. As indicated by Tabs. 13 and 14, the inner products for any $V, W \in T_I \mathcal{S}_{++}^n$ are

$$\text{LEM, AIM, and PEM} : \langle V, W \rangle^{(\alpha,\beta)}, \tag{99}$$

$$\text{LCM} : \langle \lfloor V \rfloor + \frac{1}{2}\mathbb{V}, \lfloor W \rfloor + \frac{1}{2}\mathbb{W} \rangle, \tag{100}$$

$$\text{BWM} : \frac{1}{4}\langle V, W \rangle \tag{101}$$

The above comes from the following.

(1) Eq. (99) comes from $\log_{*,I}(V) = V$ and $\mathrm{P}_{\theta*,I}(V) = \theta V$;
(2) Eq. (100) comes from $\mathrm{Chol}_{*,I}(V) = \lfloor V \rfloor + \frac{1}{2}\mathbb{V}$;
(3) Eq. (101) comes from $\mathcal{L}_I[V] = \frac{1}{2}V$.

As shown by Thanwerdas and Pennec (2023, Thm.2.1), $F_{\sqrt{\alpha+n\beta},\sqrt{\alpha}} : \{\mathcal{S}^n, \langle \cdot, \cdot \rangle^{(\alpha,\beta)}\} \to \{\mathcal{S}^n, \langle \cdot, \cdot \rangle\}$ is the linear isometry pulling the standard inner product back to the $\mathrm{O}(n)$-invariant one:

$$F_{\sqrt{\alpha+n\beta},\sqrt{\alpha}}(X) = \sqrt{\alpha}X + \frac{\sqrt{\alpha+n\beta} - \sqrt{\alpha}}{n}\,\mathrm{tr}(X)I_n, \forall X \in \mathcal{S}^n. \tag{102}$$

Given any $Y \in \mathcal{S}^n$, its inverse map is

$$
\begin{aligned}
\left(F_{\sqrt{\alpha+n\beta},\sqrt{\alpha}}\right)^{-1}(Y) &= \frac{1}{\sqrt{\alpha}}\left\{Y - \left(\frac{\sqrt{1+n\frac{\beta}{\alpha}}-1}{n}\frac{1}{\sqrt{1+n\frac{\beta}{\alpha}}}\right)\mathrm{tr}(Y)I\right\} \\
&= \frac{1}{\sqrt{\alpha}}\left\{Y - \frac{1}{n}\left(1 - \frac{1}{\sqrt{1+n\frac{\beta}{\alpha}}}\right)\mathrm{tr}(Y)I\right\} \\
&= \frac{1}{\sqrt{\alpha}}Y - \frac{1}{n}\left(\frac{1}{\sqrt{\alpha}} - \frac{1}{\sqrt{\alpha+n\beta}}\right)\mathrm{tr}(Y)I.
\end{aligned}
\tag{103}
$$

The standard orthonormal bases over the Euclidean spaces $\{\mathcal{S}^n, \langle \cdot, \cdot \rangle\}$ and $\{\mathcal{L}^n, \langle \cdot, \cdot \rangle\}$ are

$$\{\mathcal{S}^n, \langle \cdot, \cdot \rangle\} : U_{ij}^{\mathrm{sym}} = \begin{cases} E_{ii}, & \text{if } i = j, \\ \frac{E_{ij}+E_{ji}}{\sqrt{2}}, & \text{if } i > j. \end{cases} \tag{104}$$

$$\{\mathcal{L}^n, \langle \cdot, \cdot \rangle\} : U_{ij}^{\mathrm{tril}} = E_{ij}, \forall i \geq j \tag{105}$$

where $i \geq j, i, j = 1, \cdots, n$, and $\{E_{ij}\}_{i,j=1}^n$ are standard basis matrices, with the $(k, l)$ element defined as

$$(E_{ij})_{kl} = \begin{cases} 1 & \text{if } k = i \text{ and } l = j, \\ 0 & \text{otherwise.} \end{cases} \tag{106}$$

The standard orthonormal bases w.r.t. Eqs. (99) to (101) are

$$\text{LEM, AIM, PEM} : U_{ij}^{(\alpha,\beta)} \overset{(1)}{=} \begin{cases} \frac{1}{\sqrt{\alpha}}E_{ii} - \frac{1}{n}\left(\frac{1}{\sqrt{\alpha}} - \frac{1}{\sqrt{\alpha+n\beta}}\right)I, & \text{if } i = j, \\ \frac{E_{ij}+E_{ji}}{\sqrt{2\alpha}}, & \text{if } i > j. \end{cases} \tag{107}$$

$$\text{LCM} : U_{ij}^{\mathrm{LC}} \overset{(2)}{=} \begin{cases} 2E_{ii}, & \text{if } i = j, \\ E_{ij}, & \text{if } i > j. \end{cases} \tag{108}$$

$$\text{BWM} : U_{ij}^{\mathrm{BW}} \overset{(3)}{=} \begin{cases} 2E_{ii}, & \text{if } i = j, \\ \sqrt{2}(E_{ij} + E_{ji}), & \text{if } i > j. \end{cases} \tag{109}$$

Here, $i \geq j, i, j = 1, \cdots, n$. The above comes from the following.

(1) $U_{ij}^{(\alpha,\beta)} = \left(F_{\sqrt{\alpha+n\beta},\sqrt{\alpha}}\right)^{-1}\left(U_{ij}^{\mathrm{sym}}\right)$, with $F_{\sqrt{\alpha+n\beta},\sqrt{\alpha}} : \mathcal{S}^n \to \mathcal{S}^n$ as the linear isometry pulling back the Frobenius inner product to the $\mathrm{O}(n)$-invariant inner product;

(2) $f^{\text{LC}}(V) = \lfloor V \rfloor + \frac{1}{2}\mathbb{V} : \mathcal{L}^n \to \mathcal{L}^n$ is the linear isometry pulling the Frobenius inner product to Eq. (100);

(3) $f^{\text{BW}}(V) = \frac{1}{2}V : \mathcal{S}^n \to \mathcal{S}^n$ is the linear isometry pulling the Frobenius inner product back to Eq. (101);

**Riemannian exponentiation:** Next, we show $\text{Exp}_I$ under different metrics

$$\text{LEM and AIM} : \text{Exp}_I(V) \overset{(1)}{=} \exp(V), \tag{110}$$

$$\text{PEM} : \text{Exp}_I(V) \overset{(2)}{=} (I + \theta V)^{\frac{1}{\theta}}, \tag{111}$$

$$\text{LCM} : \text{Exp}_I(V) \overset{(3)}{=} \left( \lfloor V \rfloor + \text{Dexp}\left(\frac{1}{2}\mathbb{V}\right) \right) \left( \lfloor V \rfloor + \text{Dexp}\left(\frac{1}{2}\mathbb{V}\right) \right)^\top, \tag{112}$$

$$\text{BWM} : \text{Exp}_I(V) \overset{(4)}{=} I + V + \frac{1}{4}V^2 = \left( I + \frac{1}{2}V \right)^2, \tag{113}$$

The above comes from the following.

(1) $\log_{*,I}(V) = V$ and $\log I = \mathbf{0}$;
(2) $\text{P}_{\theta *, I}(V) = \theta V$;
(3) $\text{Chol}_{*,I}(V) = \lfloor V \rfloor + \frac{1}{2}\mathbb{V}$;
(4) $\mathcal{L}_I[V] = \frac{1}{2}V$.

Now, we can prove the results metric by metric.

**LEM:**

$$\text{Exp}_I \left( \sum_{i,j=1,i\geq j}^m v_{ij}^{\text{LE}}(S)U_{ij}^{(\alpha,\beta)} \right)$$

$$= \exp \left( \sum_{i,j=1,i\geq j}^m \left( \log(S) - \log(P_{ij}), Z_{ij} \rangle^{(\alpha,\beta)} U_{ij}^{(\alpha,\beta)} \right) \right). \tag{114}$$

**AIM:**

$$\text{Exp}_I \left( \sum_{i,j=1,i\geq j}^m v_{ij}^{\text{AI}}(S)U_{ij}^{(\alpha,\beta)} \right)$$

$$= \exp \left( \sum_{i,j=1,i\geq j}^m \left( \langle \log(P_{ij}^{-\frac{1}{2}}SP_{ij}^{-\frac{1}{2}}), Z_{ij} \rangle^{(\alpha,\beta)} U_{ij}^{(\alpha,\beta)} \right) \right). \tag{115}$$

**PEM:**

$$\text{Exp}_I \left( \sum_{i,j=1,i\geq j}^m v_{ij}^{\text{PE}}(S)U_{ij}^{(\alpha,\beta)} \right)$$

$$= \left( I + \theta \sum_{i,j=1,i\geq j}^m \left( \frac{1}{\theta} \langle S^\theta - P_{ij}^\theta, Z_{ij} \rangle^{(\alpha,\beta)} U_{ij}^{(\alpha,\beta)} \right) \right)^{\frac{1}{\theta}} \tag{116}$$

$$= \left( I + \sum_{i,j=1,i\geq j}^m \left( \langle S^\theta - P_{ij}^\theta, Z_{ij} \rangle^{(\alpha,\beta)} U_{ij}^{(\alpha,\beta)} \right) \right)^{\frac{1}{\theta}}.$$

**LCM:**

$$\text{Exp}_I \left( \sum_{i,j=1,i\geq j}^m v_{ij}^{\text{LC}}(S)U_{ij}^{\text{LC}} \right)$$

$$= \left( \lfloor V^{\text{LC}} \rfloor + \text{Dexp}\left(\frac{1}{2}\mathbb{V}^{\text{LC}}\right) \right) \left( \lfloor V^{\text{LC}} \rfloor + \text{Dexp}\left(\frac{1}{2}\mathbb{V}^{\text{LC}}\right) \right)^\top, \tag{117}$$

with

$$
\begin{aligned}
V^{\mathrm{LC}} &= \sum_{i,j=1,i\geq j}^{m} v_{ij}^{\mathrm{LC}}(S)U_{ij}^{\mathrm{LC}} \\
&= \sum_{i,j=1,i\geq j}^{m} \left( \left\langle \lfloor K \rfloor - \lfloor L_{ij} \rfloor + \mathrm{Dlog}(\mathbb{K}\mathbb{L}_{ij}^{-1}), \lfloor Z_{ij} \rfloor + \frac{1}{2}\mathbb{Z}_{ij} \right\rangle \right) U_{ij}^{\mathrm{LC}}
\end{aligned}
\tag{118}
$$

**BWM:**

$$
\mathrm{Exp}_I \left( \sum_{i,j=1,i\geq j}^{m} v_{ij}^{\mathrm{BW}}(S)U_{ij}^{\mathrm{BW}} \right)
\tag{119}
$$
$$
= \left( I + \frac{1}{2}V^{\mathrm{BW}} \right)^2,
$$

with $V^{\mathrm{BW}}$ defined as

$$
V^{\mathrm{BW}} = \sum_{i,j=1,i\geq j}^{m} \left\{ \frac{1}{2} \left\langle (P_{ij}S)^{\frac{1}{2}} + (SP_{ij})^{\frac{1}{2}} - 2P_{ij}, \mathcal{L}_{P_{ij}}(L_{ij}Z_{ij}L_{ij}^\top) \right\rangle U_{ij}^{\mathrm{BW}} \right\}.
\tag{120}
$$

$\square$

### J.6 PROOF OF PROP. 4.4

We begin by recalling two vector structures on the SPD manifold. Next, we identify the expression for the linear homomorphisms. Finally, we present our proof.

We define a map $\phi(\cdot) : \mathcal{S}_{++}^n \to \mathcal{L}^n$ as

$$
\phi(S) = \lfloor L \rfloor + \mathrm{Dlog}(\mathbb{L}),
\tag{121}
$$

where $P = LL^\top$ is the Cholesky decomposition. For any $P, Q \in \mathcal{S}_{++}^n$ and $t \in \mathbb{R}$, the vector structures over the SPD manifold are defined as

$$
P \oplus^{\mathrm{LE}} Q = \exp(\log(P) + \log(Q))
\tag{122}
$$
$$
t \odot^{\mathrm{LE}} P = \exp(t\log(P)) = P^t
\tag{123}
$$
$$
P \oplus^{\mathrm{LC}} Q = \phi^{-1}(\phi(P) + \phi(Q))
\tag{124}
$$
$$
t \odot^{\mathrm{LC}} P = \phi^{-1}(t\phi(P)) = P^t
\tag{125}
$$

As shown by Arsigny et al. (2005); Chen et al. (2024d), $\{\mathcal{S}_{++}^n, \oplus^{\mathrm{LE}}, \odot^{\mathrm{LE}}\}$ and $\{\mathcal{S}_{++}^n, \oplus^{\mathrm{LC}}, \odot^{\mathrm{LC}}\}$ forms vector spaces. We further present the associated linear homomorphisms.

**Lemma J.2** (SPD Homomorphisms). *Given any homomorphisms*

$$
\zeta^{\mathrm{LE}}(\cdot) : \{\mathcal{S}_{++}^n, \oplus^{\mathrm{LE}}, \odot^{\mathrm{LE}}\} \to \{\mathcal{S}_{++}^m, \oplus^{\mathrm{LE}}, \odot^{\mathrm{LE}}\},
\tag{126}
$$
$$
\zeta^{\mathrm{LC}}(\cdot) : \{\mathcal{S}_{++}^n, \oplus^{\mathrm{LC}}, \odot^{\mathrm{LC}}\} \to \{\mathcal{S}_{++}^m, \oplus^{\mathrm{LC}}, \odot^{\mathrm{LC}}\},
\tag{127}
$$

*they can be expressed as*

$$
\zeta^{\mathrm{LE}} = \exp \circ g \circ \log,
\tag{128}
$$
$$
\zeta^{\mathrm{LC}} = \phi^{-1} \circ f \circ \phi,
\tag{129}
$$

*where $f : \mathcal{L}^n \to \mathcal{L}^m$ and $g : \mathcal{S}^n \to \mathcal{S}^m$ are linear homomorphisms over the Euclidean space $\mathcal{L}^n$ and $\mathcal{S}^n$, respectively.*

*Proof.* As shown by Chen et al. (2024d), $\log(\cdot)$ is the linear isomorphism from $\{\mathcal{S}_{++}^n, \oplus^{\mathrm{LE}}, \odot^{\mathrm{LE}}\}$ to the Euclidean space $\mathcal{S}^n$ and $\phi$ is the linear isomorphism from $\{\mathcal{S}_{++}^n, \oplus^{\mathrm{LC}}, \odot^{\mathrm{LC}}\}$ to the Euclidean space $\mathcal{L}^n$. Therefore, any linear homomorphisms over these two linear spaces have the following forms:

$$
\zeta^{\mathrm{LE}} = \log^{-1} f \circ \log,
\tag{130}
$$
$$
\zeta^{\mathrm{LC}} = \phi^{-1} g \circ \phi,
\tag{131}
$$

where $f : \mathcal{S}^n \to \mathcal{S}^m$ and $g : \mathcal{L}^n \to \mathcal{L}^m$ are linear homomorphisms over the Euclidean space $\mathcal{S}^n$ and $\mathcal{L}^n$, respectively. $\square$

With all the above theoretical preparation, we begin to present our proof.

*Proof.* Given an SPD matrix $S \in \mathcal{S}_{++}^n$, Eq. (130) can be rewritten as

$$
\begin{aligned}
\zeta^{\text{LE}}(S) &\overset{(1)}{=} \exp\left( \sum_{i,j=1,i\geq j}^m \langle \log(S), A_{ij} \rangle U_{ij}^{\text{sym}} \right) \\
&\overset{(2)}{=} \exp\left( \sum_{i,j=1,i\geq j}^m \langle \log(S), A_{ij} \rangle U_{ij}^{(1,0)} \right) \\
&\overset{(3)}{=} \mathcal{F}^{\text{LE}}(S; \mathbf{A}, \mathbf{I})
\end{aligned}
\tag{132}
$$

where $\mathbf{A} = \{A_{ij} \in \mathcal{S}^n\}_{i,j=1,i\geq j}^m$ and $\mathbf{I} = \{I, \cdots, I\}$. The above comes from the following.

(1) The linear map $f$ can be represented by $\{A_{ij} \in \mathcal{S}^n\}_{i,j=1,i\geq j}^m$ under the bases $\{U_{ij}^{\text{sym}}\}_{i,j=1,i\geq j}^n$ over $\mathcal{S}^n$ and $\{U_{ij}^{\text{sym}}\}_{i,j=1,i\geq j}^m$ over $\mathcal{S}^m$;

(2) $\{U_{ij}^{\text{sym}}\}_{i,j=1,i\geq j}^m = \{U_{ij}^{(1,0)}\}_{i,j=1,i\geq j}^m$;
(3) $\text{Exp}_I = \exp$ under LEM.

Following the above logic, we have the following for $\{\mathcal{S}_{++}^n, \oplus^{\text{LC}}, \odot^{\text{LC}}\}$:

$$
\begin{aligned}
\zeta^{\text{LC}}(S) &\overset{(1)}{=} \phi^{-1}\left( \sum_{i,j=1,i\geq j}^m \langle \phi(S), A_{ij} \rangle U_{ij}^{\text{tril}} \right) \\
&\overset{(2)}{=} \mathcal{F}^{\text{LC}}(S; \mathbf{Z}, \mathbf{I}),
\end{aligned}
\tag{133}
$$

where $A_{ij} \in \mathcal{L}^n$ for $i, j = 1, \cdots, m, i \geq j$, $\mathbf{Z} = \{Z_{ij} = A_{ij} + \mathbb{D}(A_{ij}) \in \mathcal{L}^n\}_{i,j=1,i\geq j}^m$ and $\mathbf{I} = \{I, \cdots, I\}$. The above comes from the following.

(1) The linear map $g$ can be represented by $\{A_{ij}\}_{i,j=1,i\geq j}^m$;
(2) Eq. (7) and $v_{ij}^{\text{LC}}$.

$\square$

## J.7 Proof of Thm. 4.5

Before presenting our proof, we first discuss some basic facts about the ONB Grassmannian FC layer.

As implied by Eq. (30), any tangent vector $V \in T_{I_{p,n}} \text{Gr}(p, n)$ can be expressed as

$$
V = \begin{pmatrix} \mathbf{0} \\ I_{n-p} \end{pmatrix} B_V = \begin{pmatrix} \mathbf{0} \\ B_V \end{pmatrix}, \text{ with } B_V \in \mathbb{R}^{(n-p) \times p}.
\tag{134}
$$

According to Thm. 3.3 and Eq. (134), the ONB Grassmannian FC layer $\mathcal{F}(\cdot) : \text{Gr}(p, n) \to \text{Gr}(q, m)$ has the following form:

$$
Y = \text{Exp}_{I_{q,m}}\left( \sum_{\substack{i=1,\cdots,m-q \\ j=1,\cdots,m}} \left( \langle \text{Log}_{P_{ij}}(X), A_{ij} \rangle_{P_{ij}} U_{ij} \right) \right),
\tag{135}
$$

where $\{U_{ij}\}$ are the orthonormal bases over $T_{I_{q,m}} \text{Gr}(q, m)$. As discussed in Sec. 3.3, we model the FC parameters by parallel transport and Riemannian exponential map:

$$
A_{ij} = \Gamma_{I_{p,n} \to P_{ij}}(Z_{ij}),
\tag{136}
$$
$$
P_{ij} = \text{Exp}_{I_{p,n}}(\gamma_{ij}[Z_{ij}]),
\tag{137}
$$

where $Z_{ij} = \begin{pmatrix} \mathbf{0} \\ B_{Z_{ij}} \end{pmatrix} \in T_{I_{p,n}} \text{Gr}(p, n)$. Therefore, we can model each $P_{ij}$ and $A_{ij}$ by $B_{Z_{ij}} \in \mathbb{R}^{(n-p) \times p}$ and $\gamma_{ij} \in \mathbb{R}$. With the above ingredient, we present the proof in the following.

*Proof.* **The standard orthonormal basis:** As the inner product over $T_{I_{q,m}} \mathrm{Gr}(q, m)$ is the Frobenius matrix inner product (Bendokat et al., 2024, Eq. 3.2), the standard orthonormal basis over $T_{I_{q,m}} \mathrm{Gr}(q, m)$ is

$$U_{ij} = \begin{pmatrix} \mathbf{0} \\ E_{ij} \end{pmatrix}, 1 \le i \le m - q \wedge 1 \le j \le q, \tag{138}$$

where $\{E_{ij}\}$ are standard basis matrices over $\mathbb{R}^{(m-q) \times q}$

**The Riemannian exponential map at the origin:** The SVD of $V \in T_{I_{p,n}} \mathrm{Gr}(p, n)$ can be calculated via the SVD of $B_V$:

$$V = \begin{pmatrix} \mathbf{0} \\ B_V \end{pmatrix} = \begin{pmatrix} \mathbf{0} \\ O \end{pmatrix} \Sigma R^\top = \begin{pmatrix} \mathbf{0} \\ O\Sigma R^\top \end{pmatrix}, \tag{139}$$

where $B_V \overset{\mathrm{SVD}}{:=} O\Sigma R^\top$. Therefore, the Riemannian exponential map at $I_{p,n}$ can be simplified as

$$\begin{aligned}
\mathrm{Exp}_{I_{p,n}}(V) &= \begin{pmatrix} I_p \\ \mathbf{0} \end{pmatrix} R\cos(\Sigma)R^T + \begin{pmatrix} \mathbf{0} \\ O \end{pmatrix} \sin(\Sigma)R^T \\
&= \begin{pmatrix} R\cos(\Sigma)R^T \\ O\sin(\Sigma)R^T \end{pmatrix}
\end{aligned} \tag{140}$$

$v_{ij}(U)$ **under the ONB perspective:** The ONB parallel transport can be further simplified. Given $P \in \mathrm{Gr}(p, n)$, we have the following for the Riemannian logarithm

$$\mathrm{Log}_{I_{p,n}}(P) = \begin{pmatrix} \mathbf{0} \\ B_P \end{pmatrix} \overset{\mathrm{SVD}}{:=} \begin{pmatrix} \mathbf{0} \\ O_P\Sigma_P R_P^\top \end{pmatrix}, \tag{141}$$

with $B_P \overset{\mathrm{SVD}}{:=} O_P\Sigma_P R_P^\top$. For $P \in \mathrm{Gr}(p, n)$ and $Z \in T_{I_{p,n}} \mathrm{Gr}(p, n)$, the parallel transport can be further simplified:

$$\begin{aligned}
&\Gamma_{I_{p,n} \to P}(Z) \\
&= \left( \left( \begin{pmatrix} I_{p,n}R_P & \begin{pmatrix} \mathbf{0} \\ O_P \end{pmatrix} \end{pmatrix} \right) \begin{pmatrix} -\sin(\Sigma_P) \\ \cos(\Sigma_P) \end{pmatrix} \begin{pmatrix} \mathbf{0} \\ O_P \end{pmatrix}^T + \left( I - \begin{pmatrix} \mathbf{0} \\ O_P \end{pmatrix} \begin{pmatrix} \mathbf{0} \\ O_P \end{pmatrix}^T \right) \right) Z \\
&= \left( \left( -\begin{pmatrix} I_p \\ \mathbf{0} \end{pmatrix} R_P \sin(\Sigma_P) + \begin{pmatrix} \mathbf{0} \\ O_P \end{pmatrix} \cos(\Sigma_P) \right) \begin{pmatrix} \mathbf{0} \\ O_P \end{pmatrix}^T + \begin{pmatrix} I_p & \mathbf{0} \\ \mathbf{0} & I_{n-p} - O_P O_P^\top \end{pmatrix} \right) Z \\
&= \left( \begin{pmatrix} -R_P \sin(\Sigma_P) \\ O_P \cos(\Sigma_P) \end{pmatrix} \begin{pmatrix} \mathbf{0} & O_P^\top \end{pmatrix} + \begin{pmatrix} I_p & \mathbf{0} \\ \mathbf{0} & I_{n-p} - O_P O_P^\top \end{pmatrix} \right) Z \\
&= \left( \begin{pmatrix} \mathbf{0} & -R_P \sin(\Sigma_P)O_P^\top \\ \mathbf{0} & O_P \cos(\Sigma_P)O_P^\top \end{pmatrix} + \begin{pmatrix} I_p & \mathbf{0} \\ \mathbf{0} & I_{n-p} - O_P O_P^\top \end{pmatrix} \right) Z \\
&= \begin{pmatrix} I_p & -R_P \sin(\Sigma_P)O_P^\top \\ \mathbf{0} & I_{n-p} + O_P \cos(\Sigma_P)O_P^\top - O_P O_P^\top \end{pmatrix} Z \\
&= \begin{pmatrix} I_p & -R_P \sin(\Sigma_P)O_P^\top \\ \mathbf{0} & I_{n-p} + O_P \cos(\Sigma_P)O_P^\top - O_P O_P^\top \end{pmatrix} \begin{pmatrix} \mathbf{0} \\ B_Z \end{pmatrix} \\
&= \begin{pmatrix} -R_P \sin(\Sigma_P)O_P^\top B_Z \\ \left(O_P \cos(\Sigma_P)O_P^\top + I_{n-p} - O_P O_P^\top\right) B_Z \end{pmatrix}.
\end{aligned}$$

Combining all the above results, one can directly obtain the results. $\square$

## J.8 PROOF OF THM. 4.6

*Proof.* Firstly, $v_{ij}(X)$ over the Grassmannian $\widetilde{\mathrm{Gr}}(p, n)$ takes the following form:

$$\begin{aligned}
v_{ij}(X) &= \left\langle \mathrm{Log}_{P_{ij}}(X), \Gamma_{\widetilde{I}_{p,n} \to P_{ij}}(Z_{ij}) \right\rangle_{P_{ij}} \\
&\overset{(1)}{=} \frac{1}{2} \left\langle \mathrm{Log}_{P_{ij}}(X), \Gamma_{\widetilde{I}_{p,n} \to P_{ij}}(Z_{ij}) \right\rangle
\end{aligned} \tag{142}$$

where (1) comes from Tab. 15. Here, each $Z_{ij} \in T_{\widetilde{I}_{p,n}} \widetilde{\mathrm{Gr}}(p,n)$ and $P_{ij} \in \widetilde{\mathrm{Gr}}(p,n)$.

**Riemannian logarithm.** As shown by Nguyen et al. (2024, Prop. 3.12), the PP Grassmannian logarithm can be calculated by the ONB logarithm:

$$\mathrm{Log}_P^{\mathrm{PP}}(X) = \pi_{*,\pi(P)}\left(\mathrm{Log}_{\pi^{-1}(P)}^{\mathrm{ONB}}(\pi^{-1}(X))\right), \tag{143}$$

where $\pi(U) = UU^\top : \mathrm{Gr}(p,n) \to \widetilde{\mathrm{Gr}}(p,n)$ is the Riemannian isometry, and $\pi_{*,U}(V) = UV^\top + VU^\top$ is the differential map for all $U \in \mathrm{Gr}(p,n)$ and $V \in T_U \mathrm{Gr}(p,n)$.

**Tangent vector and Riemannian exponential map at the identity.** As implied by Eq. (32), any tangent vector at the identity has the following form:

$$V = \begin{pmatrix} 0 & B^T \\ B & 0 \end{pmatrix} \in T_{\widetilde{I}_{p,n}} \widetilde{\mathrm{Gr}}(p,n) \text{ with } B \in \mathbb{R}^{(n-p)\times p}. \tag{144}$$

The Riemannian exponential at the identity can also be simplified:

$$\mathrm{Exp}_{\widetilde{I}_{p,n}}(V) = \exp([V, \widetilde{I}_{p,n}])\widetilde{I}_{p,n} \exp(-[V, \widetilde{I}_{p,n}])$$

$$= \exp\left(\begin{pmatrix} 0 & -B^T \\ B & 0 \end{pmatrix}\right) \widetilde{I}_{p,n} \exp\left(\begin{pmatrix} 0 & -B^T \\ B & 0 \end{pmatrix}\right)^\top \tag{145}$$

$$= \left(\exp\left(\begin{pmatrix} 0 & -B^T \\ B & 0 \end{pmatrix}\right)\right)_{1:p} \left(\left(\exp\left(\begin{pmatrix} 0 & -B^T \\ B & 0 \end{pmatrix}\right)\right)_{1:p}\right)^\top$$

with $(\cdot)_{1:p}$ as the first-$p$ columns of the input square matrix.

**Parallel transport starting at the identity.** The parallel transport along geodesic from $\widetilde{I}_{p,n}$ to $P \in \widetilde{\mathrm{Gr}}(p,n)$ can also be simplified. For any $V \in T_{\widetilde{I}_{p,n}} \widetilde{\mathrm{Gr}}(p,n)$, denoting $\bar{P} = \mathrm{Log}_{\widetilde{I}_{p,n}}(P)$, we have the following:

$$\Gamma_{\widetilde{I}_{p,n} \to P}(V) \overset{(1)}{=} \exp\left(\left[\bar{P}, \widetilde{I}_{p,n}\right]\right) V \exp\left(-\left[\bar{P}, \widetilde{I}_{p,n}\right]\right)$$

$$\overset{(2)}{=} \exp\left(\begin{pmatrix} 0 & -B_P^T \\ B_P & 0 \end{pmatrix}\right) V \exp\left(\begin{pmatrix} 0 & -B_P^T \\ B_P & 0 \end{pmatrix}\right)^\top \tag{146}$$

The above derivation comes from the following.

(1) Tab. 15;

(2) $\bar{P} = \begin{pmatrix} 0 & B_P^T \\ B_P & 0 \end{pmatrix}$

**Trivialization and simplification** Combining Eqs. (142) and (144) to (146), we model each $P_{ij}$ such that

$$P_{ij} = \exp\left(\begin{pmatrix} 0 & -B_{P_{ij}}^T \\ B_{P_{ij}} & 0 \end{pmatrix}\right) \widetilde{I}_{p,n} \exp\left(\begin{pmatrix} 0 & -B_{P_{ij}}^T \\ B_{P_{ij}} & 0 \end{pmatrix}\right)^\top \tag{147}$$

where $B_{P_{ij}} = \gamma_{ij}[B_{Z_{ij}}]$ with $Z_{ij} = \begin{pmatrix} 0 & B_{Z_{ij}}^T \\ B_{Z_{ij}} & 0 \end{pmatrix}$ and $B_{Z_{ij}} \in \mathbb{R}^{(n-p)\times p}$.

Denoting $O_{ij} = \exp\left(\begin{pmatrix} 0 & -B_{P_{ij}}^T \\ B_{P_{ij}} & 0 \end{pmatrix}\right)$, $v_{ij}(X)$ can be simplified as

$$v_{ij}(X) = \frac{1}{2}\left\langle \pi_{*,\pi(P)}\left(\mathrm{Log}_{(O_{ij})_{1:p}}^{\mathrm{ONB}}(\pi^{-1}(X))\right), O_{ij} Z_{ij} O_{ij}^\top \right\rangle \tag{148}$$

**Orthonormal bases.** Finally, let us deal with the orthonormal bases over $T_{\widetilde{I}_{q,m}} \widetilde{\mathrm{Gr}}(q,m)$. For any tangent vector $V_1, V_2 \in T_{\widetilde{I}_{q,m}} \widetilde{\mathrm{Gr}}(q,m)$, we have the following:

$$\langle V_1, V_2 \rangle_{\widetilde{I}_{p,n}} = \frac{1}{2} \langle V_1, V_2 \rangle$$

$$= \frac{1}{2}\left\langle \begin{pmatrix} 0 & B_{V_1}^T \\ B_{V_1} & 0 \end{pmatrix}, \begin{pmatrix} 0 & B_{V_2}^T \\ B_{V_2} & 0 \end{pmatrix} \right\rangle \tag{149}$$

$$= \langle B_{V_1}, B_{V_2} \rangle$$

Therefore, the orthonormal bases are

$$U_{ij} = \begin{pmatrix} 0 & E_{ij}^\top \\ E_{ij} & 0 \end{pmatrix}, \forall i = 1, \cdots, m - q \wedge j = 1, \cdots, q \tag{150}$$

where $E_{ij} \in \mathbb{R}^{(m-q) \times q}$ is the standard basis matrix.

Combining Eqs. (145), (148) and (150), one can readily obtain the results. $\qquad \square$

