# OpenReview forum: "Building Transformation Layers for Riemannian Neural Networks"
_ICLR.cc/2026/Conference — ICLR 2026 Conference Withdrawn Submission_

### Official Review · Reviewer_zURU · 2025-10-27

**Soundness:** 3
**Presentation:** 3
**Contribution:** 3
**Rating:** 8
**Confidence:** 2

**Summary:**

This paper proposes a general framework for constructing Fully Connected (FC) and convolutional layers on Riemannian manifolds. The framework relies solely on fundamental Riemannian operators. The authors show that several existing manifold-specific FC layers (e.g., Poincaré FC, gyro-SPD FC layers, flat-SPD FC, Lorentz linear layers). The framework is instantiated across ten geometries, including: three hyperbolic models, five well-known SPD geometries, and both ONB and projector perspectives of the Grassmannian. The proposed framework is evaluated with several datasets: Disease, Airport, Pubmed, and Cora for hyperbolic spaces, and Radar, HD05, FPHA for SPD manifolds.

**Strengths:**

The core novelty is the unification of FC and convolutional transformations across many Riemannian spaces.
The authors broadly instantiate the proposed framework across hyperbolic, SPD, and Grassmannian manifolds. The framework behaves conysistently across these settings, suggesting general usefulness.
The authors evaluate the framework with 7 datasets in both hyperbolic spaces and SPD manifolds, that strengthens argments.

**Weaknesses:**

There are still some points not clear and need to be clarified:
1. why point-to-hyperplane distance leads to FC transformations not directly explained
2. Computational cost is not discussed. Riemannian operations can be expensive depending on the manifold and metric. A runtime or FlOPs comparison across manifolds would make the generality claim more convincing.

**Questions:**

1. why point-to-hyperplane distance leads to FC transformations
2. Can the authors provide more details on comparing computational cost, that would strenghten the generability of the propose framework.
3. Can the authors provide more details on trivialization. what paremeters are in tangent spaces, what are in the manifold.

---

### Official Review · Reviewer_6ri4 · 2025-10-31

**Soundness:** 3
**Presentation:** 3
**Contribution:** 2
**Rating:** 2
**Confidence:** 4

**Summary:**

This paper proposes a framework for constructing FC and convolutional layers for neural networks on hyperbolic, Symmetric Positive
Definite (SPD), and Grassmannian spaces. The authors show that several existing Riemannian FC and convolutional layers are special cases of their proposed layers. The authors conducted experiments on link prediction, radar classification, and action recognition to validate the proposed framework.

**Strengths:**

- Three popular Riemannian manifolds are investigated.
- The authors provide theoretical results to justify their construction of FC and convolutional layers.

**Weaknesses:**

- The paper lacks of novelty (the weakest point).
- Some experimental settings are missing (minor).
- Experiements are insuffient to demonstrate the efficacy of the proposed method (moderately important).

**Questions:**

1) This paper relies heavily on the works in [Shimizu et al., 2021; Nguyen and Yang, 2023; Nguyen et al., 2024; Chen et al., 2024c]. Here, the key idea is to build Riemannian FC and convolutional layers from Riemannian analogs of linear transformations just as Euclidean FC and convolutional layers are built from linear transformations. This idea was first introduced for hyperbolic manifold [Shimizu et al., 2021]. The main difficulty is how one can adap this idea to matrix manifolds, which was already done in [Nguyen et al., 2024]. In the present work, the authors simply apply a similar approach to obtain formulae for different metrics associated with different Riemannian manifolds (which have been throughly studied in previous works). While the authors have attempted to show that the proposed framework is applicable to a variety of Riemannian manifolds, it still stays within the context of hyperbolic and matrix manifolds (e.g., SPD and Grassmann manifolds) considered in [Ganea et al., 2018; Shimizu et al., 2021; Nguyen et al., 2024]. The paper would be much appreciated if the authors can develop a new approach for constructing Riemannian FC and convolutional layers rather than using an existing approach for different Riemannian metrics which are well-understood.

To give a bit more details of what I said above: The key ingredient to the construction of FC and convolutional layers in the approaches of [Shimizu et al., 2021; Nguyen et al., 2024] as well as in the present paper is the Riemannian point-to-hyperplane distance. In the present work, it is given in line 146. This distance coincides with those in [Nguyen and Yang, 2023] in the cases of AIM, LEM, LCM. Eq. (2) is the one proposed in [Nguyen et al., 2024] for hypergyroplane that contains the origin and is orthogonal to an axis of the output space. The construction of FC and convolutional layers for Grassmann space, while not being presented in [Nguyen et al., 2024], is quite direct from that work in my opinion (since the point-to-hyperplane distance can be derived as in that work). Therefore, the paper focuses on the same context as [Shimizu et al., 2021; Nguyen et al., 2024] while using the same idea as these works for the construction of the proposed layers.

In addition to my concern above on the novelty of the paper, I also have concern on the practical benefit of the proposed models:

- Hyperbolic space: compared to [Shimizu et al., 2021; Bdeir et al., 2024], the authors add the Klein model to their study. However, Table 4 shows that either the Poincare model or the Hyperboloid model outperforms the Klein model in all cases.

- SPD space: compared to [Nguyen et al., 2024], the authors add the PEM and BWM metrics to their study. However, Table 6 show that for 3 out of 4 datasets (e.g., HDM05, FPHA, NTU60), none of these metrics is the best performer. The only case where PEM performs the best is on Radar dataset, but the performance gap between PEM and LEM is minor (98.43 $\pm$ 0.44 vs. 98.27 $\pm$ 0.48).

That being said, by considering some Riemannian metrics and hyperbolic models that were not considered in previous works, the present paper provides new FC and convolutional layers, but these do not seem to be as effective as existing layers.

2) In Section 3.1, the authors state that "The previous work typically requires a case-by-case derivation for a specific geometry". I think this is because those derivations require closed form formulas of the exponential/logarithm maps, and this limitation applies to any methods built upon those maps, including the proposed method. So I failed to see the advantage of the propsed method compared to previous ones in this matter.

3) Experimental settings are not always clearly provided. For instance, for NTU60 dataset, the authors said "we focus on the mutual action in NTU60". However, as far as I know, there are different experimental settings for this dataset and it is not clear which one they authors used.

4) Some improvements observed from the experiemnts, in my opinion, mainly come from specific implementations and experiemental settings. This is because the proposed layers seem to follow a similar construction as existing ones, and I failed to see how notable improvements can be obtained if similar constructions are used.


Questions:

1. Could the authors explain the new contributions in terms of layers' construction w.r.t. [Shimizu et al., 2021; Nguyen and Yang, 2023; Nguyen et al., 2024; Chen et al., 2024c] ?

2. What are the challenges to address, given that the key ingredients have already been given in [Shimizu et al., 2021; Nguyen and Yang, 2023; Nguyen et al., 2024; Chen et al., 2024c] ?

3. What are the differences in the architectures of SPDNN-LEM and GyroLE in Table 6 ? How the input are computed ? How many epochs are used for each experiment ? How many runs for each model ? Some of those are briefly given in the Appendix but I would like to ask for further details and discussions.

---

### Official Review · Reviewer_E7jf · 2025-11-01

**Soundness:** 2
**Presentation:** 2
**Contribution:** 2
**Rating:** 2
**Confidence:** 4

**Summary:**

The paper studies transformation layers for Riemannian neural networks. Their construction in S3.1 starts with fully-connected (FC) layers in Euclidean space and generalizes them to the manifold case in Def 3.1/Thm 3.3, and then extends them to convolutional layers. They instantiate these layers on the hyperbolic, SPD, and Grassmannian manifolds and experimentally evaluate on link prediction, radar classification, and human action recognition tasks.

**Strengths:**

The generality of the paper is appealing: it is nice to have a general framework for parameterized transformations (e.g., FC or convolutional layers) on manifolds. The paper proposes one way of doing this and shows that is already captures existing methods specialized to other manifolds (like GyroSPD++). The experimental results also show these layers have practical value too (although in the next section I do have some concerns on the experimental results not being complete).

**Weaknesses:**

I have a few concerns on weaknesses that hold me back from recommending an accept at this point. I'm very open to discussing them throughout the review process.

The most significant is on the experimental results: setting out to be (near) state-of-the-art on the experimental settings across all of the geometries considered in the paper is as ambitious goal that I see as necessary for publishing the paper. The current experiments are lacking. For example, the link prediction results in Table 4/5 significantly underperform the reported results from Table 1 of the [Riemannian Residual Neural Networks paper](https://proceedings.neurips.cc/paper_files/paper/2023/file/c868aa7437dc9b29e674cd2e25689021-Paper-Conference.pdf) --- for example, they get 98.4 on disease and 95.2 on airport. And, Table 6 comparing to GyroSPD++ significantly underperforms the results reported in Table 1 of the [original GyroSPD++ paper](https://arxiv.org/abs/2405.19206) --- the paper under review states this is due to not being able to reproduce GyroSPD++ and not having code available, but the difference seems too significant and warrants further justification.

My other concern is on the positioning and slight over-claiming of the paper, making statements such as "this paper extends fundamental FC and convolutional layers to operate on Riemannian manifold": it does not give enough credit to other attempts at creating general Riemannian layers, such as in the Riemannian Residual Neural Networks paper, which also just uses an exponential map and projection onto the tangent space. My recommendation here would be to tone down these claims, and give more credit to previous attempts at creating general Riemannian layers.

**Questions:**

I am very open to discussing the weaknesses I have pointed out, especially on the experiments seeming incomplete.

Beyond that, I do have a larger question for discussion on the use and motivation for Riemannian layers that I am curious about. If we are modeling a mapping from manifold data for classification or regression tasks (on the simplex or Euclidean space), then at what point should the model stop using Riemannian layers and start just using the standard (Euclidean) layers used in other regression/classification tasks? It seems like there could be some representational argument for first embedding the Riemannian input data into a latent Euclidean space with Riemannian layers, and then processing that with Eucldiean layers. I could also see an argument that if the input data is already a real-valued vector space (via the representation of the points on the manifold or ambient Euclidean space), then the entire task could be processed with just Euclidean layers. Is it just experimental what transformation works the best, or is there some strong argument that the transformations need to be done on the manifold?

And one last question to discuss is on the comparison between the Riemannian layer proposed and other the general Riemannian layers, like in the Riemannian ResNet paper combining a Euclidean module with a projection onto the tangent space and exponential map. Conceptually or theoretically, is there any argument for why the proposed layer should be more capacitive or learnable? Or are there only experimental justifications? I would find this helpful to discuss (and eventually include in the paper)

---

### Official Review · Reviewer_eqsD · 2025-11-01

**Soundness:** 3
**Presentation:** 2
**Contribution:** 3
**Rating:** 4
**Confidence:** 4

**Summary:**

The paper proposes a framework for developing fully connected (FC) layers for Riemannian neural networks. Performance is evaluated across multiple manifolds, and the accuracy seems to outperform state-of-the-art methods including GyroSPD++ and RResNet.

**Strengths:**

Fully connected layers for Riemannian neural networks have been a design problem for some time, and this paper seems to have a novel solution to it that works well in practice. The accuracy numbers in Table 5 and Table 6 are in particular quite good.

**Weaknesses:**

Much of the rhetoric in the introduction seems to be overclaiming. There are a lot of previous FC layers that the paper cites but doesn't discuss in any detail. No real explanation is given for how the proposed FC setup differs from these previous approaches. And in particular the setup of RResNet (which is mentioned in the experiments section as a basis for the experiments, but not in the intro) seems to satisfy all the desiderata: it uses only geometric Riemannian operators and works generally across manifolds. Some more reason why the proposed FC method works better would improve the paper.

**Questions:**

Can this Riemannian FC layer truly be said to be based on Riemannian operators if it involves taking a particular basis at "the origin"? Many manifolds do not have an obvious origin. Is this method somehow independent of which point is chosen as an origin.

---

### Note · Authors · 2025-11-14

I have read and agree with the venue's withdrawal policy on behalf of myself and my co-authors.